# DEEP GRAPH-LEVEL ORTHOGONAL HYPERSPHERE COMPRESSION FOR ANOMALY DETECTION

## ABSTRACT

Graph-level anomaly detection aims to identify abnormal samples of a set of graphs in an unsupervised manner. It is non-trivial to find a reasonable decision boundary between normal data and anomalous data without using any anomalous data in the training stage, especially for data in graphs. This paper first proposes a novel deep graph-level anomaly detection model, which learns the graph representation with maximum mutual information between substructure features and global structure features while exploring a hypersphere anomaly decision boundary. We implement an orthogonal projection layer to keep the training data distribution consistent with the decision hypersphere thus avoiding erroneous evaluations. More importantly, we further propose projecting the normal data into the interval region between two co-centered hyperspheres, which makes the normal data distribution more compact and effectively overcomes the issue of outliers falling close to the center of the hypersphere. The numerical and visualization results on a few graph datasets demonstrate the effectiveness and superiority of our methods in comparison to many baselines and state-of-the-art.

## 1 INTRODUCTION

Anomaly detection is an essential task with various applications, such as detecting abnormal patterns or actions in credit-card fraud, medical diagnosis, sudden natural disasters (Aggarwal, 2017), etc. Usually, in anomaly detection, the training data only contain normal data and are used to train a model that can distinguish unusual patterns from abnormal ones. Anomaly detection on tabular data and images has been extensively studied recently (Ruff et al., 2018; Goyal et al., 2020; Chen et al., 2022; Liznerski et al., 2021; Sohn et al., 2021). In contrast, there is little work on graph data despite the fact that graph data anomaly detection is very useful in various problems, such as identifying abnormal communities in social networks or detecting unusual protein structures in biology experiments. Compared with the other types of data, graph data is inherently complicated and rich in structural and relational information. The complexity of graph structure facilitates us to learn graph-level representations with discriminative patterns in many supervised tasks (e.g., graph classification). As for graph-level anomaly detection, however, the intricate graph structure brings many obstacles to this unsupervised learning problem.

Graph anomaly detection usually composes four families: anomalous edge (Ouyang et al., 2020; Xu et al., 2020), node (Zhu & Zhu, 2020; Bojchevski & Günnemann, 2018), sub-graph (Wang et al., 2018; Zheng et al., 2018), and graph-level detections (Zheng et al., 2019; Chalapathy et al., 2018). Herein, the target of the graph-level algorithms is to explore a regular group pattern and distinguish the abnormal manifestations of the group. Group abnormal behaviors usually foreshadow some unusual events and thus play an important role in practical applications. In the past five years, few approaches have focused on graph-level anomaly detection because of the difficulty of representing graphs into feature vectors without using any label information. Graph kernel can measure the similarity between graphs and regard the result as a representation non-strictly or implicitly. Based on this, graph anomaly detection task usually performs as two-stage. In our experiments (see Section 4), we also find that one-class SVM with graph kernels sometimes yields unsatisfying performances since graph kernels may not be effective enough to quantify the similarity between graphs. So there is a large room for improvement regarding graph anomaly detection to our best knowledge.

Concerning end-to-end models, Ma et al. (2022) proposed a global and local knowledge distillation method for graph-level anomaly detection, which learns rich global and local normal pattern information by random joint distillation of graph and node representations. The method needs to train two GCNs jointly at a high time cost. Zhao & Akoglu (2021) combined the Deep SVDD objective function and graph isomorphism network to learn a hypersphere of normal samples. Qiu et al. (2022) also sought a hypersphere decision boundary and optimized the representations learned by $k$ GNNs close to the reference GNN while maximizing the differences between $k$ GNNs, but did not consider the relationship between the graph-level representation and node features. Collecting all approaches based on the hypersphere assumption in graph anomaly detection, we find that the practical decision region may be an ellipsoid instead of a standard hypersphere, thus causing the error when the standard hypersphere evaluation is employed. Except for that, our experiment also confirms that anomalous data may appear in decision regions that are not filled with normal data, especially near the center of the hypersphere.

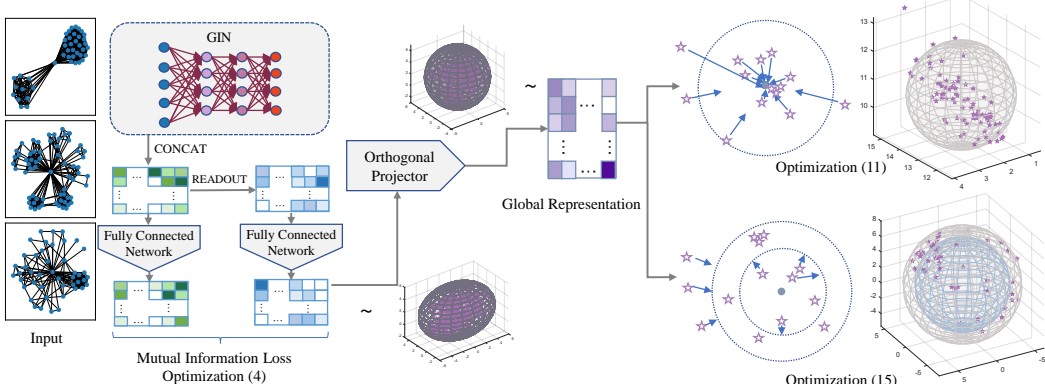

Figure 1: Architecture of the proposed models *(right top: DOHSC; right bottom: DO2HSC)*.

In order to effectively explore a better representation without label information and obtain a more suitable decision boundary with high efficiency, in this paper, we propose a one-class deep graph-level anomaly detection method and its improved version. The first proposed model, **D**eep **O**rthogonal **H**ypersphere **C**ontraction (DOHSC), uses the mutual information of local feature maps and the global representation to learn a high-quality representation and simultaneously optimizes it to distribute in a hypersphere area. An orthogonal projection layer then renders the decision region more hyperspherical and compact to decrease evaluation errors. With regard to phenomenon that anomalous data falling close to the hyperspherical center, an improved graph-level **D**eep **O**rthogonal **Bi**-**H**yper**s**phere **C**ompression (DO2HSC) for anomaly detection architecture is proposed. From a cross-sectional point of view, DO2HSC limits the decision area (of normal data) to an interval enclosed by two co-centered hyperspheres and learns the orthogonality-projected representation similarly. The framework of the methods mentioned above is shown in Figure 1 correspondingly. Furthermore, we define a new evaluation way according to DO2HSC, and comprehensive experimental results verify the effectiveness of all proposed methods. In summary, the main contributions of our work are listed as follows.

- First, we present a new graph-level hypersphere contraction algorithm for anomaly detection tasks, which is jointly trained via mutual information loss between local and global representations and hypersphere decision loss.

- Second, we impose an orthogonal projection layer on the proposed model to promote training data distribution close to the standard hypersphere, thus avoiding errors arising from inconsistencies between assessment criteria and actual conditions.

- Finally, we propose an improved graph-level deep orthogonal bi-hypersphere compression model to further explore a decision region enclosed by two co-centered hyperspheres, which can effectively prevent anomalous data falling close to the hyperspherical center and surpass baselines significantly in the experiments.

## 2 PROPOSED APPROACH

In this section, we first introduce a joint learning architecture in detail, named as Graph-Level Deep Orthogonal Hypersphere Contraction. Then an improved algorithm is proposed to compensate for the underlying assumption's deficiency.

### 2.1 GRAPH-LEVEL DEEP ORTHOGONAL HYPERSPHERE CONTRACTION

#### 2.1.1 VALLINA MODEL

Given a set of graphs $\mathbb{G} = \{G_1, ..., G_N\}$ with $N$ samples, the proposed model aims to learn a $k$-dimensional representation and then set a soft-boundary according to it. In this paper, the Graph Isomorphism Network (GIN) (Xu et al., 2019) is employed to obtain the graph representation in three stages: first, input the graph data and integrate neighbors of the current node (AGGREGATE); second, combine neighbor and current node features (CONCAT); finally, integrate all node information (READOUT) into one global representation. Mathematically, the $i$-th node features of $l$-th layer and the global features of its affiliated $j$-th graph would be denoted as

$$
\begin{aligned}
\mathbf{z}_\Phi^i &= \text{CONCAT}(\{\mathbf{z}_i^{(l)}\}_{l=1}^L), \\
\mathbf{Z}_\Phi(G_j) &= \text{READOUT}(\{\mathbf{z}_\Phi^i\}_{i=1}^{|G_j|}),
\end{aligned}
\tag{1}
$$

where $\mathbf{z}_\Phi^i \in \mathbb{R}^{1 \times k}$ and $\mathbf{Z}_\Phi(G_j) \in \mathbb{R}^{1 \times k}$. To integrate the contained information and enhance the differentiation between node-level and global-level representations, we append additional fully connected layers denoted as the forms $M_\Upsilon(\cdot)$ and $T_\Psi(\cdot)$, respectively, where $\Upsilon$ and $\Psi$ are the parameters of the added layers. So the integrated node-level and graph-level representations are obtained via

$$
\begin{aligned}
\mathbf{h}_{\Phi,\Upsilon}^i &:= M_\Upsilon(\mathbf{z}_\Phi^i), \\
\mathbf{H}_{\Phi,\Psi}(G_j) &:= T_\Psi(\mathbf{Z}_\Phi(G_j)),
\end{aligned}
\tag{2}
$$

To better capture the local information, we utilize the batch optimization property of neural networks to maximize the mutual information (MI) between local and global representations in each batch $\mathbf{G} \subseteq \mathbb{G}$, which is defined by Sun et al. (2020) as the following term:

$$
\hat{\Phi}, \hat{\Psi}, \hat{\Upsilon} = \arg\max_{\Phi, \Psi, \Upsilon} I_{\Phi,\Psi,\Upsilon}\left(\mathbf{h}_{\Phi,\Upsilon}, \mathbf{H}_{\Phi,\Psi}(\mathbf{G})\right).
\tag{3}
$$

Specifically, the mutual information estimator $I_{\Phi,\Psi,\Upsilon}$ follows Jensen-Shannon MI estimator (Nowozin et al., 2016) with a positive-negative sampling method as below,

$$
\begin{aligned}
I_{\Phi,\Psi,\Upsilon}\left(\mathbf{h}_{\Phi,\Upsilon}, \mathbf{H}_{\Phi,\Psi}(\mathbf{G})\right) :=& \sum_{G_j \in \mathbf{G}} \frac{1}{|G_j|} \sum_{u \in G_j} I_{\Phi,\Psi,\Upsilon}\left(\mathbf{h}_{\Phi,\Upsilon}^u(G_j), \mathbf{H}_{\Phi,\Psi}(\mathbf{G})\right) \\
=& \sum_{G_j \in \mathbf{G}} \frac{1}{|G_j|} \sum_{u \in G_j} \Big[ \mathbb{E}\big(-\sigma\left(-\mathbf{h}_{\Phi,\Upsilon}^u(\mathbf{x}^+) \times \mathbf{H}_{\Phi,\Psi}(\mathbf{x})\right)\big) \\
& \qquad\qquad\qquad - \mathbb{E}\big(\sigma\left(\mathbf{h}_{\Phi,\Upsilon}^u(\mathbf{x}^-) \times \mathbf{H}_{\Phi,\Psi}(\mathbf{x})\right)\big)\Big],
\end{aligned}
\tag{4}
$$

where a softplus function $\sigma(z) = \log(1 + e^z)$ is activated after vector multiplication between node and graph representations. For $\mathbf{x}$ as an input sample graph, we calculate the expected mutual information with its positive samples $\mathbf{x}^+$ and negative samples $\mathbf{x}^-$, which are generated from distribution across all graphs in a subset. Given each $G = (\mathcal{V}_G, \mathcal{E}_G)$ and node set $\mathcal{V}_G = \{v_i\}_{i=1}^{|G|}$, the positive and negative samples are divided in this way:

$$
\mathbf{x}^+ = \begin{cases} \mathbf{x}_{ij}, & \text{if } v_i \in G_j, \\ 0, & \text{otherwise.} \end{cases}
\tag{5}
$$

And $\mathbf{x}^-$ produces the opposite result in each of the conditions above. In the next step, a data-enclosing decision boundary is required for our anomaly detection task. According to the assumption that most normal data can locate in a hypersphere, the center of this decision boundary should

be initialized through

$$\mathbf{c} = \frac{1}{N} \sum_{i=1}^{N} \mathbf{H}_{\Phi,\Psi}(G_i). \tag{6}$$

With this center, we expect to optimize the learned representation of normal data to be distributed as close to it as possible, so that the unexpected anomalous data falling out of this hypersphere would be detected. Besides, the regularization term is adopted to avoid over-fitting problems. Collectively denote the weight parameters of $\Phi$, $\Psi$ and $\Upsilon$ as $\mathcal{Q} := \Phi \cup \Psi \cup \Upsilon$, we formulate the training loss with two joint objectives – Hypersphere Contraction and MI:

$$\min_{\Phi,\Psi,\Upsilon} \frac{1}{|\mathbf{G}|} \sum_{i=1}^{|\mathbf{G}|} \|\mathbf{H}_{\Phi,\Psi}(G_i) - \mathbf{c}\|^2 + \lambda I_{\Phi,\Psi,\Upsilon}\left(\mathbf{h}_{\Phi,\Upsilon}, \mathbf{H}_{\Phi,\Psi}(\mathbf{G})\right) + \frac{\mu}{2} \sum_{\mathbf{Q} \in \mathcal{Q}} \|\mathbf{Q}\|_F^2, \tag{7}$$

where $|\mathbf{G}|$ denotes the number of graphs in batch $\mathbf{G}$ and $\lambda$ is a trade-off factor, the third term is a network weight decay regularizer with the hyperparameter $\mu$.

### 2.1.2 ORTHOGONAL PROJECTION LAYER

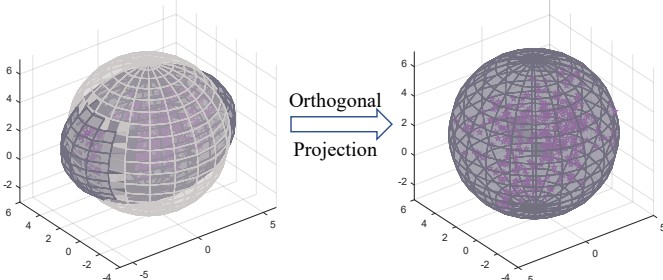

Figure 2: Variations in decision boundary with and without the orthogonal projection layer. In the left subfigure, the real decision region where training data are distributed may be an ellipsoid (dark grey). This contradicts with the hypersphere decision boundary (light grey) set by Optimization 7. After orthogonal projection, the ellipsoid is expected to be transformed into a standard hypersphere, which avoids the evaluation error. *Note: here data are simulated only for illustration.*

However, an empirical study shows that a hyperellipsoid is commonly observed during deep representation learning. This phenomenon would lead to inaccuracies in the final test because the evaluation results are based on the hypersphere decision region. Problem (7) obviously cannot guarantee the soft-boundary of learned representation to be a standard hypersphere like Figure 2. Accordingly, we append an orthogonal projection layer after obtaining the global representation. Note that we pursue orthogonal features of latent representation rather than computing the projection onto the column or row space of $\mathbf{H}_{\Phi,\Psi}$. This method is equivalent to performing PCA and using the standardized principal components. Our experiments also justify the necessities of this projection step and standardization process, which will be discussed further in Section 4.4 and Appendix G. Specifically, the projection layer can be formulated as

$$\tilde{\mathbf{H}}_{\Phi,\Psi,\Theta}(G) = \mathrm{Proj}_{\Theta}(\mathbf{H}_{\Phi,\Psi}(G)) = \mathbf{H}_{\Phi,\Psi}\mathbf{W}, \quad \text{subject to} \quad \tilde{\mathbf{H}}_{\Phi,\Psi,\Theta}^{\top}\tilde{\mathbf{H}}_{\Phi,\Psi,\Theta} = \mathbf{I}_{k'} \tag{8}$$

where $\Theta := \{\mathbf{W} \in \mathbb{R}^{k \times k'}\}$ are the projection parameters, $\mathbf{I}_{k'}$ denotes an identity matrix, and $k'$ is the projected dimension.

Note that to achieve (8), one may consider adding a regularization term $\frac{\alpha}{2}\|\tilde{\mathbf{H}}_{\Phi,\Psi,\Theta}^{\top}\tilde{\mathbf{H}}_{\Phi,\Psi,\Theta} - \mathbf{I}_{k'}\|_F^2$ with large enough $\alpha$ to the objective, which is not very effective and will lead to one more tuning hyperparameter. Instead, we propose to achieve (8) via singular value decomposition, i.e.,

$$\begin{aligned} \mathbf{U}\mathbf{\Lambda}\mathbf{V}^{\top} &= \mathbf{H}_{\Phi,\Psi}, \\ \mathbf{W} &:= \mathbf{V}_{k'}\mathbf{\Lambda}_{k'}^{-1}, \end{aligned} \tag{9}$$

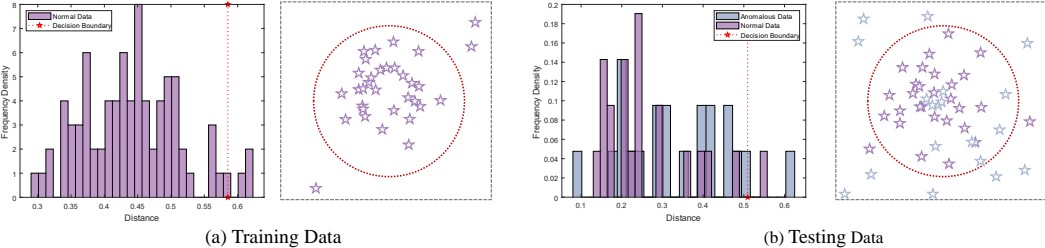

(a) Training Data                                      (b) Testing Data

Figure 3: Illustration of inevitable flaws in DOHSC on both the training and testing data of COX2. Left: the $\ell_2$-norm distribution of 4-dimensional distances learned from the real dataset; Right: the pseudo-layout in two-dimensional space sketched by reference to the empirical distribution.

where $\mathbf{\Lambda} = \mathrm{diag}(\rho_1, \rho_2, ..., \rho_{|\mathbf{G}|})$ and $\mathbf{V}$ are the diagonal matrix with sigular values and right-singular matrix of $\mathbf{H}_{\Phi,\Psi}$, respectively. It needs to be emphasized that $\mathbf{V}_{k'} := [\mathbf{v}_1, ..., \mathbf{v}_{k'}]$ denotes the first $k'$ right-singular vectors and $\mathbf{\Lambda}_{k'} := \mathrm{diag}(\rho_1, ..., \rho_{k'})$. In each forward propagation epoch, the original weight parameter is substituted to a new matrix $\mathbf{W}$ in subsequent loss computations.

### 2.1.3 ANOMALY DETECTION

Attaching with an orthogonal projection layer, the improved initialization of the center is rewritten in the following form

$$\tilde{\mathbf{c}} = \frac{1}{N} \sum_{i=1}^{N} \tilde{\mathbf{H}}_{\Phi,\Psi,\Theta}(G_i) \tag{10}$$

and the final objective function for anomaly dectection tasks in a mini-batch would become

$$\min_{\Theta,\Phi,\Psi,\Upsilon} \frac{1}{|\mathbf{G}|} \sum_{i=1}^{|\mathbf{G}|} \|\tilde{\mathbf{H}}_{\Phi,\Psi,\Theta}(G_i) - \tilde{\mathbf{c}}\|^2 + \lambda \sum_{\mathbf{G} \in \mathbb{G}} I_{\Phi,\Psi,\Upsilon} \left( \mathbf{h}_{\Phi,\Upsilon}, \tilde{\mathbf{H}}_{\Phi,\Psi,\Theta}(\mathbf{G}) \right) + \frac{\mu}{2} \sum_{\mathbf{Q} \in \mathcal{Q}} \|\mathbf{Q}\|_F^2. \tag{11}$$

After the training stage, a decision boundary $\hat{r}$ will be fixed, which is calculated based on the $1 - \nu$ percentile of the training data distance distribution:

$$\hat{r} = \arg\min_{r} \mathcal{P}(\mathbf{D} \leq r) \geq \nu \tag{12}$$

where $\mathbf{D} := \{d_i\}_{i=1}^{N}$ follows a sampled distribution $\mathcal{P}$, and $d_i = \|\tilde{\mathbf{H}}_{\Phi,\Phi,\Theta}(G_i) - \tilde{\mathbf{c}}\|$. Accordingly, the anomalous score of $i$-th instance is defined as follows:

$$s_i = d_i^2 - \hat{r}^2 \tag{13}$$

where $\mathbf{s} = (s_1, s_2, \ldots, s_N)$. It is evident that when the score is positive, the instance is identified as abnormal, and the opposite is considered normal.

The detailed procedures of algorithm are summarized into Algorithm 1 (see Appendix A). It starts with graph representation learning and promotes the training data to approximate the center of a hypersphere while adding an orthogonal projection layer. Unfortunately, it can be observed from Figure 3 that the anomalous data would appear in partial regions of the learned decision hypersphere, which are not filled by the training data, especially the region close to the center. To handle this particular situation, an improved graph-level anomaly detection approach, termed as Graph-Level Deep Orthogonal Bi-Hypersphere Compression, will be designed in the next section.

### 2.2 GRAPH-LEVEL DEEP ORTHOGONAL BI-HYPERSPHERE COMPRESSION

As Figure 3 suggests, we found peculiar phenomena in our empirical results that the learned distribution of training data sometimes could not satisfy the hypersphere assumption, where anomalous data might appear within the decision region and therefore led to suboptimal detection performance. To explore the reason behind, we examine the counter-intuitive behaviors of high-dimensional Gaussian

distributions, and the simulation results imply the *soap-bubble* problem, where anomalous samples could exist near the center of learned hypersphere (see Appendix B for more details). Since DOHSC cannot detect anomalies close to the center, we propose an improved approach, which sets the decision boundary as an interval region between two co-centered hyperspheres. This can narrow the decision area's scope and induce normal data to fill the entire interval area as much as possible.

After the same graph representation learning stage, we firstly utilize the DOHSC model for a few epochs and initialize the large radius $r_{\max}$ and the small radius $r_{\min}$ of the interval area according to the $1 - \nu$ percentile and $\nu$ of the sample distances distribution, respectively. The aforementioned descriptions can be denoted mathematically as below.

$$
\begin{aligned}
r_{\max} &= \arg \min_r \mathcal{P}(\mathbf{D} \leq r) \geq \nu, \\
r_{\min} &= \arg \min_r \mathcal{P}(\mathbf{D} \leq r) \geq 1 - \nu.
\end{aligned}
\tag{14}
$$

After fixing the decision boundaries $r_{max}$ and $r_{min}$, the improved training loss is also set with a trade-off factor $\lambda$, which implicitly emphasizes the importance of the max-min term:

$$
\begin{aligned}
\min_{\Theta, \Phi, \Psi, \Upsilon} \frac{1}{|\mathbf{G}|} \sum_{i=1}^{|\mathbf{G}|} \left( \max\{d_i, r_{max}\} - \min\{d_i, r_{min}\} \right) &+ \lambda \sum_{\mathbf{G} \in \mathbb{G}} I_{\Phi, \Psi, \Upsilon} \left( \mathbf{h}_{\Phi, \Upsilon}, \tilde{\mathbf{H}}_{\Phi, \Psi, \Theta}(\mathbf{G}) \right) \\
&+ \frac{\mu}{2} \sum_{\mathbf{Q} \in \mathcal{Q}} \|\mathbf{Q}\|_F^2.
\end{aligned}
\tag{15}
$$

This decision loss has the lowest bound $r_{max} - r_{min}$ and can be jointly minimized with mutual information effectively. Besides, the evaluation standard of test data is also needed to change based on this interval structure. More specifically, all instances located in the inner hypersphere and out of the outer hypersphere should be identified as anomalous graphs; only those located in the interval area should be regarded as normal data. Compared with equation 13, we reset a new score function to award the positive samples beyond $[r_{min}, r_{max}]$ and meanwhile punishing the negative samples within the range. Accordingly, the distinctive scores are calculated by

$$
s_i = (d_i - r_{max}) \cdot (d_i - r_{min}),
\tag{16}
$$

where $i \in \{1, ..., N\}$. This way, we can also effectively identify a sample's abnormality by its score. In general, the improved deep graph-level anomaly detection algorithm changes the decision boundary and effectively makes the normal area more compact. Correspondingly, the new practical evaluation is raised to adapt to the improved detection way. Eventually, we summarize the detailed procedures of the optimization into Algorithm 2 (see Appendix A).

## 3 CONNECTION WITH PREVIOUS WORK

Actually, few studies have been undertaken in graph-level anomaly detection (GAD). Existing solutions to GAD tasks can be categorized into two types: a two-stage approach and an end-to-end one. Two-stage GAD methods first transform graphs into graph embeddings by graph neural networks or into similarities between graphs by graph kernels, and then apply off-the-shelf anomaly detectors such as local outlier factor (LOF) (Breunig et al., 2000), one-class support vector machine (OCSVM) (Schölkopf et al., 1999), etc. The drawback of the two-stage method is that the graph feature extractor and outlier detector are independent, and some graph kernels produce "hand-crafted" features that are deterministic without much space to improve. End-to-end approaches overcome this problem by utilizing deep graph learning techniques, such as graph convolutional network (GCN) (Welling & Kipf, 2016) and graph isomorphism network (GIN) (Xu et al., 2019). With an anomaly measure as the objective, end-to-end approaches jointly learn an effective graph representation for GAD task (Zhao & Akoglu, 2021; Qiu et al., 2022; Ma et al., 2022). Zhao & Akoglu (2021) optimized one-class model based on deep support vector data description (Deep SVDD) as the anomaly measure. Here we clarify the differences between (Zhao & Akoglu, 2021) and ours. First, the proposed model employed mutual information loss in the graph learning stage to obtain the graph representation incorporating local and global information. On the contrary, Zhao & Akoglu (2021) directly utilized the readout result of GIN. We also impose an orthogonal projection on the learned representation to maintain consistency between the learned decision boundary and normal data distribution. More importantly, we present a new approach to constructing decision boundary. The

new approach learns two hyperspheres, between which the region accommodates normal data and hence yields more space for abnormal data. As for graph kernel, we summarized the previous work in Appendix C.

## 4 EXPERIMENTS

### 4.1 DATASET

In this work, we test our method on six real-world graph datasets[1], which contain three social networks datasets (COLLAB, COX2, and IMDB-Binary) and three bioinformatics datasets (DD, ER_MD, and MUTAG). The details of the datasets are shown in Table 1.

Table 1: Description for six datasets.

| Datasets | # Graphs | Avg. # Nodes | Avg. # Edges | # Classes | # Graph Labels |
|---|---|---|---|---|---|
| COLLAB | 5000 | 74.49 | 2457.78 | 3 | 2600 / 775 / 1625 |
| COX2 | 467 | 42.43 | 44.54 | 2 | 365 / 102 |
| ER_MD | 446 | 21.33 | 234.85 | 2 | 265 / 181 |
| MUTAG | 188 | 17.93 | 19.79 | 2 | 63 / 125 |
| DD | 1178 | 284.32 | 715.66 | 2 | 691 / 487 |
| IMDB-Binary | 1000 | 19.77 | 96.53 | 2 | 500 / 500 |

### 4.2 BASELINES

We compare our method with the following unsupervised graph-level anomaly detection methods: Random Walk (RW) (Gärtner et al., 2003; Kashima et al., 2003), Shortest Path Kernel (SP) (Borgwardt & Kriegel, 2005), Weisfeiler-Lehman Sub-tree Kernel (WL) (Shervashidze et al., 2011), Neighborhood Hash Kernel (NH) (Hido & Kashima, 2009). Besides graph kernels, we also compare three graph-level representation learning methods: Deep One Class Model with GIN network (OCGIN) (Zhao & Akoglu, 2021), Graph-level embedding Learning via Mutual Information Maximization+Deep SVDD (infoGraph+Deep SVDD) (Sun et al., 2020; Ruff et al., 2018), Global and Local Knowledge Distillation for Graph-level Anomaly Detection (GLocalKD) (Ma et al., 2022) and One Class Graph Transformation Learning (OCGTL) (Qiu et al., 2022).

### 4.3 RESULTS

In this section, extensive experimental results are displayed to validate the effectiveness of the proposed models. The averages and standard deviations of the Area Under Operating Characteristic Curve (AUC) are used to support the comparable experiments by repeating each algorithm ten times. The higher value of the AUC metric represents better performance. Tables 2–4 report the AUC metric and its standard deviations. It can be seen that the proposed methods basically achieve the best AUC values compared to other algorithms on all datasets. Both approaches outperform other state-of-the-art baselines, and DO2HSC obtains superior performance and get $5\%$ higher performance than other algorithms on many datasets, such as MUTAG, COLLAB Class 1, and ER_MD Class 0, IMDB-Binary Class 1, etc. It is woth mentioning that we defeat infoGraph+Deep SVDD with a large improvement, which is a degraded version of the proposed models, thus showing that promoting representation learning towards the anomaly detection goal is meaningful and well targeted.

The anomaly detection visualization results of DO2HSC are displayed in Figure 4 and those of DOHSC are also shown in Appendix F. We draw them by setting the projection dimension to three directly. Results of different perspectives are given to avoid blind spots in the field of vision, demonstrating excellent performance. Hence, it can be concluded that the effect of the improved model is in line with our motivation and shows much potential.

---

[1]https://ls11-www.cs.tu-dortmund.de/staff/morris/graphkerneldatasets

Table 2: Average AUCs with standard deviation (10 trials) of different graph-level anomaly detection algorithms. We assess models by regarding every data class as normal data, respectively. The best results are marked in **bold** and '–' means out of memory.

| | COLLAB | | | COX2 | |
|---|---|---|---|---|---|
| | 0 | 1 | 2 | 0 | 1 |
| SP+OCSVM | $0.5910 \pm 0.0000$ | $0.8397 \pm 0.0000$ | $0.7902 \pm 0.0000$ | $0.5408 \pm 0.0000$ | $0.5760 \pm 0.0000$ |
| $WL_2$+OCSVM | $0.5051 \pm 0.0000$ | $0.7989 \pm 0.0000$ | $0.6977 \pm 0.0000$ | $0.5736 \pm 0.0000$ | $0.4286 \pm 0.0000$ |
| $WL_5$+OCSVM | $0.5079 \pm 0.0000$ | $0.8021 \pm 0.0000$ | $0.7884 \pm 0.0000$ | $0.5990 \pm 0.0000$ | $0.4376 \pm 0.0000$ |
| $WL_8$+OCSVM | $0.5106 \pm 0.0000$ | $0.8035 \pm 0.0000$ | $0.7953 \pm 0.0000$ | $0.5979 \pm 0.0000$ | $0.5057 \pm 0.0000$ |
| $WL_{10}$+OCSVM | $0.5122 \pm 0.0000$ | $0.8031 \pm 0.0000$ | $\mathbf{0.7996 \pm 0.0000}$ | $0.5937 \pm 0.0000$ | $0.5034 \pm 0.0000$ |
| NH+OCSVM | $0.5976 \pm 0.0000$ | $0.8054 \pm 0.0000$ | $0.6414 \pm 0.0000$ | $0.4841 \pm 0.0000$ | $0.4717 \pm 0.0000$ |
| RW+OCSVM | – | – | – | $0.5243 \pm 0.0000$ | $0.6553 \pm 0.0000$ |
| OCGIN | $0.4217 \pm 0.0606$ | $0.7565 \pm 0.2035$ | $0.1906 \pm 0.0857$ | $0.5964 \pm 0.0578$ | $0.5683 \pm 0.0768$ |
| infoGraph+Deep SVDD | $0.5662 \pm 0.0597$ | $0.7926 \pm 0.0986$ | $0.4062 \pm 0.0978$ | $0.4825 \pm 0.0624$ | $0.5029 \pm 0.0700$ |
| GLocalKD | $0.4638 \pm 0.0003$ | $0.4330 \pm 0.0016$ | $0.4792 \pm 0.0004$ | $0.3861 \pm 0.0131$ | $0.3143 \pm 0.0383$ |
| OCGTL | $0.6504 \pm 0.0433$ | $0.8908 \pm 0.0239$ | $0.4029 \pm 0.0541$ | $0.5541 \pm 0.032$ | $0.4862 \pm 0.0224$ |
| DOHSC (Ours) | $\mathbf{0.9185 \pm 0.0455}$ | $\mathbf{0.9755 \pm 0.0030}$ | $0.5450 \pm 0.0469$ | $0.6263 \pm 0.0333$ | $\mathbf{0.6805 \pm 0.0168}$ |
| DO2HSC (Ours) | $0.6718 \pm 0.0353$ | $0.9153 \pm 0.0070$ | $0.7188 \pm 0.0260$ | $\mathbf{0.6329 \pm 0.0292}$ | $0.6518 \pm 0.0481$ |

Table 3: Average AUCs with standard deviation (10 trials) of different graph-level anomaly detection algorithms. We assess models by regarding every data class as normal data, respectively. The best results are marked in **bold**.

| | ER_MD | | MUTAG | |
|---|---|---|---|---|
| | 0 | 1 | 0 | 1 |
| SP+OCSVM | $0.4092 \pm 0.0000$ | $0.3824 \pm 0.0000$ | $0.5917 \pm 0.0000$ | $0.2608 \pm 0.0000$ |
| $WL_2$+OCSVM | $0.3702 \pm 0.0000$ | $0.3262 \pm 0.0000$ | $0.5976 \pm 0.0000$ | $0.2960 \pm 0.0000$ |
| $WL_5$+OCSVM | $0.4446 \pm 0.0000$ | $0.2933 \pm 0.0000$ | $0.6509 \pm 0.0000$ | $0.2352 \pm 0.0000$ |
| $WL_8$+OCSVM | $0.4571 \pm 0.0000$ | $0.3199 \pm 0.0000$ | $0.4556 \pm 0.0000$ | $0.2176 \pm 0.0000$ |
| $WL_{10}$+OCSVM | $0.4297 \pm 0.0000$ | $0.2761 \pm 0.0000$ | $0.5089 \pm 0.0000$ | $0.2048 \pm 0.0000$ |
| NH+OCSVM | $0.5155 \pm 0.0200$ | $0.3648 \pm 0.0000$ | $0.7959 \pm 0.0274$ | $0.1679 \pm 0.0062$ |
| RW+OCSVM | $0.4820 \pm 0.0000$ | $0.3484 \pm 0.0000$ | $0.8698 \pm 0.0000$ | $0.1504 \pm 0.0000$ |
| OCGIN | $0.5645 \pm 0.0323$ | $0.4358 \pm 0.0538$ | $0.8491 \pm 0.0424$ | $0.4933 \pm 0.1525$ |
| infoGraph+Deep SVDD | $0.5312 \pm 0.1545$ | $0.5682 \pm 0.0704$ | $0.8805 \pm 0.0448$ | $0.6166 \pm 0.2052$ |
| GLocalKD | $0.5781 \pm 0.1790$ | $0.7154 \pm 0.0000$ | $0.3952 \pm 0.2258$ | $0.2965 \pm 0.2641$ |
| OCGTL | $0.2755 \pm 0.0317$ | $0.6915 \pm 0.0207$ | $0.6570 \pm 0.0210$ | $0.7579 \pm 0.2212$ |
| DOHSC (Ours) | $0.6620 \pm 0.0308$ | $0.5184 \pm 0.0793$ | $0.8822 \pm 0.0432$ | $0.8115 \pm 0.0279$ |
| DO2HSC (Ours) | $\mathbf{0.6867 \pm 0.0226}$ | $\mathbf{0.7351 \pm 0.0159}$ | $\mathbf{0.9089 \pm 0.0609}$ | $\mathbf{0.8250 \pm 0.0790}$ |

Table 4: Average AUCs with standard deviation (10 trials) of different graph-level anomaly detection algorithms. We assess models by regarding every data class as normal data, respectively. The best results are marked in **bold** and '–' means out of memory.

| | DD | | IMDB-Binary | |
|---|---|---|---|---|
| | 0 | 1 | 0 | 1 |
| SP+OCSVM | $0.6856 \pm 0.0000$ | $0.4474 \pm 0.0000$ | $0.4592 \pm 0.0000$ | $0.4716 \pm 0.0000$ |
| $WL_2$+OCSVM | $0.6225 \pm 0.0000$ | $0.4946 \pm 0.0000$ | $0.4294 \pm 0.0000$ | $0.4607 \pm 0.0000$ |
| $WL_5$+OCSVM | $0.7127 \pm 0.0000$ | $0.4666 \pm 0.0000$ | $0.4940 \pm 0.0000$ | $0.4243 \pm 0.0000$ |
| $WL_8$+OCSVM | $0.7343 \pm 0.0000$ | $0.4455 \pm 0.0000$ | $0.5066 \pm 0.0000$ | $0.4143 \pm 0.0000$ |
| $WL_{10}$+OCSVM | $0.7397 \pm 0.0000$ | $0.4302 \pm 0.0000$ | $0.5157 \pm 0.0000$ | $0.4115 \pm 0.0000$ |
| NH+OCSVM | $\mathbf{0.7424 \pm 0.0000}$ | $0.3684 \pm 0.0000$ | $0.5321 \pm 0.0000$ | $0.4652 \pm 0.0000$ |
| RW+OCSVM | – | – | $0.4951 \pm 0.0000$ | $0.5311 \pm 0.0000$ |
| OCGIN | $0.6659 \pm 0.0444$ | $0.6003 \pm 0.0534$ | $0.4047 \pm 0.1083$ | $0.3332 \pm 0.0649$ |
| infoGraph+Deep SVDD | $0.3942 \pm 0.0436$ | $0.6484 \pm 0.0236$ | $0.6353 \pm 0.0277$ | $0.5836 \pm 0.0995$ |
| GLocalKD | $0.1952 \pm 0.0000$ | $0.2203 \pm 0.0001$ | $0.5383 \pm 0.0124$ | $0.4812 \pm 0.0101$ |
| OCGTL | $0.6990 \pm 0.0260$ | $0.6767 \pm 0.0280$ | $0.6510 \pm 0.0180$ | $0.6412 \pm 0.0127$ |
| DOHSC (Ours) | $0.7083 \pm 0.0188$ | $0.7579 \pm 0.0154$ | $\mathbf{0.6609 \pm 0.0033}$ | $\mathbf{0.7705 \pm 0.0045}$ |
| DO2HSC (Ours) | $0.7320 \pm 0.0194$ | $\mathbf{0.7651 \pm 0.0317}$ | $0.6406 \pm 0.0642$ | $0.7101 \pm 0.0429$ |

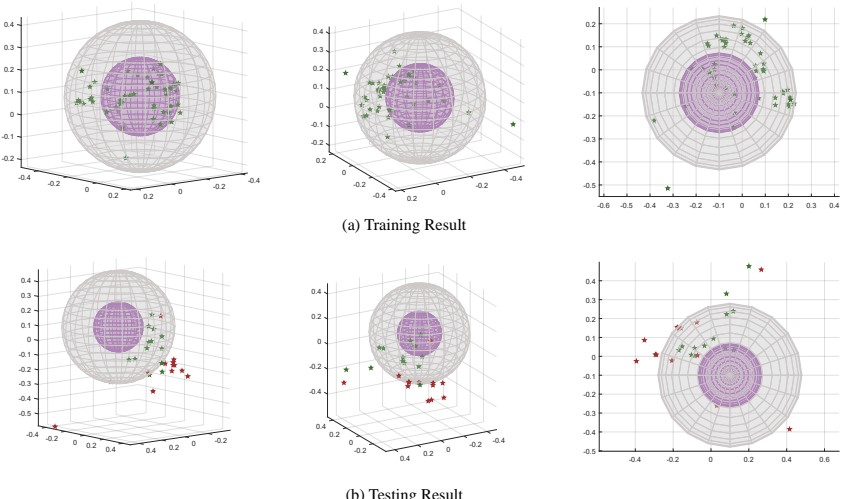

(a) Training Result

(b) Testing Result

Figure 4: Visualization results of the DO2HSC on MUTAG Class 0 in different perspectives.

## 4.4 ABLATION STUDY

In this section, we display the ablation study of the orthogonal projection layer on three datasets. From quantitive comparisons, we concluded that orthogonality positively influences all performance to some extent. It also well supports our assumption discussed in Section 2.1.2.

Table 5: Ablation study of the orthogonal projection layer. We test models by regarding every data class as normal data, respectively. The best performance is highlighted in **bold**.

|  | Orthogonal Projection Layer | Class | MUTAG | COX2 | IMDB-Binary |
|---|---|---|---|---|---|
| DOHSC | × | 0 | 0.8521 | 0.5817 | 0.5880 |
|  |  | 1 | 0.8912 | 0.5737 | 0.6082 |
|  | ✓ | 0 | **0.8639** | **0.6433** | **0.6192** |
|  |  | 1 | **0.9088** | **0.6077** | **0.6830** |
| DO2HSC | × | 0 | 0.8934 | 0.6281 | 0.6252 |
|  |  | 1 | 0.7504 | 0.6178 | 0.6541 |
|  | ✓ | 0 | **0.9467** | **0.7158** | **0.6933** |
|  |  | 1 | **0.9184** | **0.7074** | **0.6700** |

Except for the aforementioned contents, please see the detailed experiment configurations, parameter sensitivity and robustness of the proposed models, supplemented visualizations of distance distributions for anomaly detection and visualization comparison between proposed models with orthogonal projection layer and without orthogonal projection layer in Appendix G, which can also strongly support our theory and validate the effectiveness.

## 5 CONCLUSION

This paper has proposed two novel end-to-end graph-level AD methods, DOHSC and DO2HSC, which combined the effectiveness of mutual information between node-level and global features to learn graph representation and the power of hypersphere compression. DOHSC and DO2HSC mitigate the possible shortcomings of hypersphere boundary learning by applying an orthogonal projection for global representation. Furthermore, DO2HSC projects normal data between the interval area of two co-centered hyperspheres to significantly alleviate the *soap-bubble* issue. Our comprehensive experimental results strongly demonstrate the superiority of DOHSC and DO2HSC on multifarious datasets. In future work, we will explore more efficient and excellent graph representations and refine our learning process for decision regions.

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

## A  Supplemented Algorithm

Algorithm 1 summarizes the procedure of DOHSC in detail. It starts with graph representation learning and promotes the training data to approximate the center of a hypersphere while adding an orthogonal projection layer. Also, DO2HSC is recapped in Algorithm 2 and also starts with same graph representation learning. Differently, DOHSC is utilized of few epochs to initial the decision boundaries and after that, improved optimization is applied.

---

**Algorithm 1** Graph-Level Deep Orthogonal Hypersphere Contraction (DOHSC)

---

**Input:** The input graph set $\mathbb{G}$, dimensions of GIN hidden layers $k$ and orthogonal projection layer $k'$, a trade-off parameter $\lambda$ and the coefficient of regularization term $\mu$, pretraining epoch $\mathcal{T}$, learning rate $\eta$.
**Output:** The anomaly detection scores $\mathbf{s}$.
 1: Initialize the network parameters $\Phi$, $\Psi$, $\Upsilon$ and the orthogonal projection layer parameter $\Theta$;
 2: **for** $t \to \mathcal{T}$ **do**
 3:     **for** each batch $\mathbf{G}$ **do**
 4:         Calculate $I_{\Phi,\Psi,\Upsilon}\left(\mathbf{h}_{\Phi,\Upsilon}, \mathbf{H}_{\Phi,\Psi}(\mathbf{G})\right)$ via equation 4;
 5:         Back-propagation GIN and update $\Phi$, $\Psi$ and $\Upsilon$, respectively;
 6:     **end for**
 7: **end for**
 8: Update the orthogonal parameter $\Theta$ of orthogonal projection layer by equation 9;
 9: Obtain the global orthogonal latent representation by equation 8;
10: Initialize the center of hypersphere by equation 10;
11: **repeat**
12:     **for** each batch $\mathbf{G}$ **do**
13:         Calculate total loss via 11;
14:         Back-propagation and update $\Phi$, $\Psi$, $\Upsilon$ and $\Theta$, respectively;
15:         Further update the orthogonal parameter $\Theta$ of orthogonal projection layer by equation 9;
16:     **end for**
17: **until** convergence
18: Compute decision boundary $r$ by equation 12;
19: Calculate the anomaly detection scores $\mathbf{s}$ through equation 13;
20: **return** The anomaly detection scores $\mathbf{s}$.

---

## B  Related Proof of Bi-Hypersphere Learning Motivation

The traditional idea of detecting outliers is to inspect the distribution's tails with an ideal assumption that the normal data are Gaussian. Following the assumption, one may argue that an anomalous sample can be distinguished by its large Euclidean distance to the data center ($\ell_2$ norm $\|\mathbf{z} - \mathbf{c}\|$, where $\mathbf{c}$ denotes the centroid), and accordingly, the abnormal dataset is $\{\mathbf{z} : \|\mathbf{z} - \mathbf{c}\| > r\}$ for some decision boundary $r$. However in high dimensional space, Gaussian distributions look like soap-bubble [2], which means the normal data are more likely to locate in a bi-hypersphere(Vershynin, 2018), such as $\{\mathbf{z} : r_{min} < \|\mathbf{z} - \mathbf{c}\| < r_{max}\}$. To better understand this counterintuitive behavior, let us generate some normal samples $\mathbf{X} \sim \mathcal{N}(\mathbf{0}, \mathbf{I}_d)$, where $d$ is the data dimension in $\{1, 10, 50, 100, 200, 500\}$. As Figure 6 indicates, only the univariate Gaussian has a near-zero mode, whereas other high-dimensional Gaussian distributions leave plenty of offcenter spaces in blank. The soap-bubble problem in high-dimensional distributions is well demonstrated in Table 6: the higher the dimension is, the greater the quantities of data further away from the center, especially for 0.01-quantile distance. Thus, we cannot make a sanguine assumption that **all** of the normal data locate within some radius of a hypersphere (i.e. $\{\mathbf{z} : \|\mathbf{z} - \mathbf{c}\| < r\}$). Using Lemma 1 of (Laurent & Massart, 2000), we can prove that proposition 1, which matches the values in the Table 6 that when the dimension is larger, normal data are more likely lies away from center.

**Proposition 1** *Suppose $\mathbf{z}_1, \mathbf{z}_2, \cdots, \mathbf{z}_n$ are sampled from $\mathcal{N}(\mathbf{0}, \mathbf{I}_d)$ independently. Then for any $\mathbf{z}_i$ and all $t \geq 0$, the following inequality holds:*

---

[2]https://www.inference.vc/high-dimensional-gaussian-distributions-are-soap-bubble/

---

**Algorithm 2** Graph-Level Deep Orthogonal Bi-Hypersphere Compression (DO2HSC)

---

**Input:** The input graph set $\mathbb{G}$, dimensions of GIN hidden layers $k$ and orthogonal projection layer $k'$, a trade-off parameter $\lambda$ and the coefficient of regularization term $\mu$, pretraining epoch $\mathcal{T}_1$, iterations of initializing decision boundaries $\mathcal{T}_2$, learning rate $\eta$.

**Output:** The anomaly detection scores $\mathbf{s}$.

    Initialize the network parameters $\Phi$, $\Psi$, $\Upsilon$ and the orthogonal projection layer parameter $\Theta$;

2:  **for** $t \rightarrow \mathcal{T}_1$ **do**

      **for** each batch **G do**

4:        Calculate $I_{\Phi,\Psi,\Upsilon}\left(\mathbf{h}_{\Phi,\Upsilon}, \mathbf{H}_{\Phi,\Psi}(\mathbf{G})\right)$ via equation 4;

        Back-propagation GIN and update $\Phi$, $\Psi$ and $\Upsilon$, respectively;

6:    **end for**

    **end for**

8:  Update the orthogonal parameter $\Theta$ of orthogonal projection layer by equation 9;

    Obtain the global orthogonal latent representation by equation 8;

10: Initialize the center of hypersphere by equation 10;

    **for** $t \rightarrow \mathcal{T}_2$ **do**

12:    **for** each batch **G do**

        Calculate total loss via 11;

14:        Back-propagation and update $\Phi$, $\Psi$, $\Upsilon$ and $\Theta$, respectively;

        Further update the orthogonal parameter $\Theta$ of orthogonal projection layer by equation 9;

16:    **end for**

    **end for**

18: Initialize decision boundaries $r_{max}$ and $r_{min}$ via equation 14;

    **repeat**

20:    **for** each batch **G do**

        Calculate the improved total loss via 15;

22:        Back-propagation and update $\Phi$, $\Psi$, $\Upsilon$ and $\Theta$, respectively;

        Further update the orthogonal parameter $\Theta$ of orthogonal projection layer by equation 9;

24:    **end for**

    **until** convergence

26: Calculate the anomaly detection scores $\mathbf{s}$ through equation 16;

    **return** The anomaly detection scores $\mathbf{s}$.

---

$$\mathbb{P}\left[\|\mathbf{z}_i\| \geq \sqrt{d - 2\sqrt{dt}}\right] \geq 1 - e^{-t}.$$

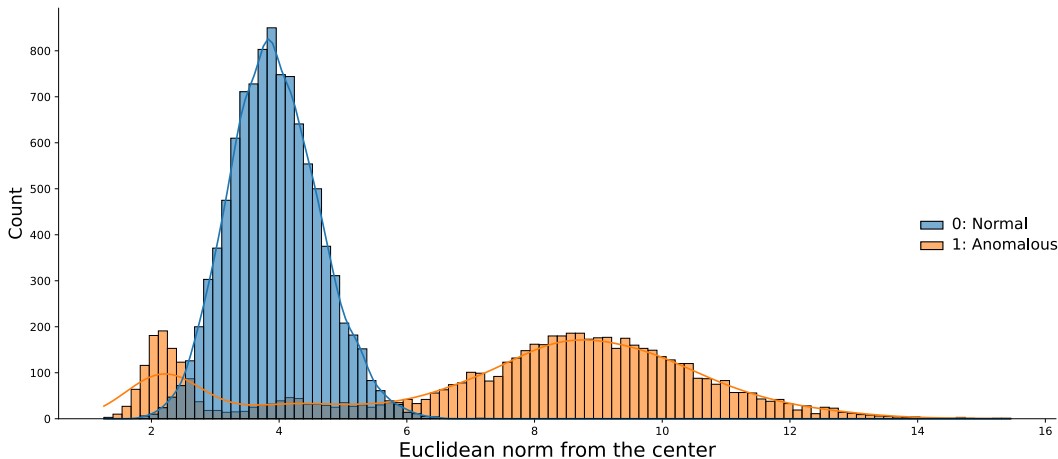

Figure 5: Histogram of distances (Euclidean norm) from the center of normal samples under 16-dimensional Gaussian distributions $\mathcal{N}(\mathbf{0}, \mathbf{I})$. Three groups of anomalous data are also 16-dimensional and respectively sampled from $\mathcal{N}(\mu_1, \frac{1}{10}\mathbf{I})$, $\mathcal{N}(\mu_2, \mathbf{I})$, and $\mathcal{N}(\mu_3, 5\mathbf{I})$, where the population means $\mu_1, \mu_2, \mu_3$ are randomized within $[0, 1]$ for each dimension.

Table 6: Offcenter distance under multivariate Gaussian at different dimensions and quantiles.

| Quantile (correspond to $r_{min}$) | dim=1 | dim=10 | dim=50 | dim=100 | dim=200 | dim=500 |
|---|---|---|---|---|---|---|
| 0.01 | 0.0127 | 1.5957 | 5.5035 | 8.3817 | 12.5117 | 20.6978 |
| 0.25 | 0.3115 | 2.5829 | 6.5380 | 9.4908 | 13.6247 | 21.8542 |
| 0.50 | 0.6671 | 3.0504 | 7.0141 | 9.9662 | 14.1054 | 22.3337 |
| 0.75 | 1.1471 | 3.5399 | 7.5032 | 10.4386 | 14.5949 | 22.8200 |
| 0.99 | 2.5921 | 4.8265 | 8.7723 | 11.6049 | 15.7913 | 24.0245 |

We also simulate a possible case of outlier detection, in which data are all sampled from 16-dimensional Gaussian with orthogonal covariance: 10,000 normal samples follow $\mathcal{N}(\mathbf{0}, \mathbf{I})$ and the first group of 1,000 outliers are from $\mathcal{N}(\mu_1, \frac{1}{10}\mathbf{I})$, the second group of 500 outliers are from $\mathcal{N}(\mu_2, \mathbf{I})$, and the last group of 2,000 outliers are from $\mathcal{N}(\mu_3, 5\mathbf{I})$. Figure 5 well exemplifies that abnormal data from other distribution (group-1 outliers) could fall into the small distance away from the center of the normal samples.

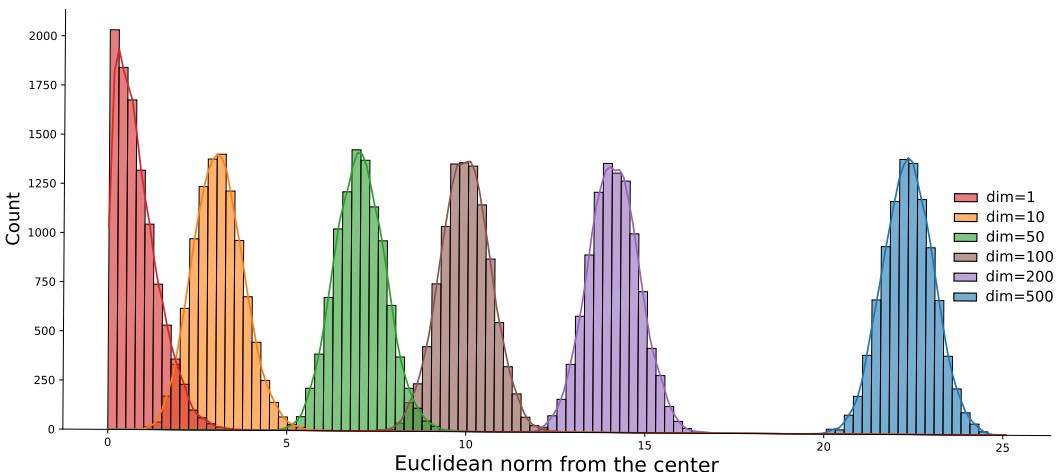

Figure 6: Histogram of distances (Euclidean norm) from the center of 10,000 random samples under (univariate or) high-dimensional Gaussian distributions $\mathcal{N}(\mathbf{0}, \mathbf{I})$.

## C  RELATED WORK ON GRAPH KERNEL

To learn with graph-structured data, graph kernels that measure the similarity between graphs become an established and widely-used approach (Kriege et al., 2020). A large body of work has emerged in the past years, including kernels based on neighborhood aggregation techniques and walks and paths. Shervashidze et al. (2011) introduced Weisfeiler-Lehman (WL) algorithm, a well-known heuristic for graph isomorphism. In Hido & Kashima (2009), Neighborhood Hash kernel was introduced, where the neighborhood aggregation function is binary arithmetic. The most influential graph kernel for paths-based kernels is the shortest-path (SP) kernel by Borgwardt & Kriegel (2005). For walks-based kernels, Gärtner et al. (2003) and Kashima et al. (2003) simultaneously proposed graph kernels based on random walks, which count the number of label sequences along walks that two graphs have in common. These graph kernel methods have the desirable property that they do not rely on the vector representation of data explicitly but access data only via the Gram matrix.

## D  EXPERIMENT CONFIGURATION

In this part, the experiment settings are listed for reproducing. First, each dataset is divided into two parts: training and testing sets. We randomly sample eighty percent of the normal data as the

training set, and the remaining normal data together with the randomly sampled abnormal data in a one-to-one ratio to form the testing set.

With regard to the experiment settings of compared graph-kernel baselines, we adopted the classical AD method, One-Class SVM (OC-SVM) (Schölkopf et al., 2001) and used 10-fold cross-validation to make a fair comparison. All graph kernels via GraKel (Siglidis et al., 2020) to extract a Kernel matrix and apply OC-SVM in scikit-learn (Pedregosa et al., 2011) are implemented. Specifically, we selected Floyd Warshall as the SP kernel's algorithm and set lambda as 0.01 for the RW kernel. WL kernel algorithm is sensitive to the number of iterations, so we test four types with the iteration of $\{2, 5, 8, 10\}$ and denote them as $WL_2$, $WL_5$, $WL_8$, $WL_{10}$. For all graph kernels, the outputs are normalized. About infoGraph+Deep SVDD, the first stage runs in 20 epochs, and the second stage pretrains 50 epochs and trains 100 epochs. In OCGIN, GLocalKD, and OCGTL, their default or reported parameter settings are adopted to reproduce experimental results. But there still exists some special situations like, due to the limited device, the relatively large-scale dataset is tested with a small batch size, such as DD. This may lead to worse performance for compared algorithms.

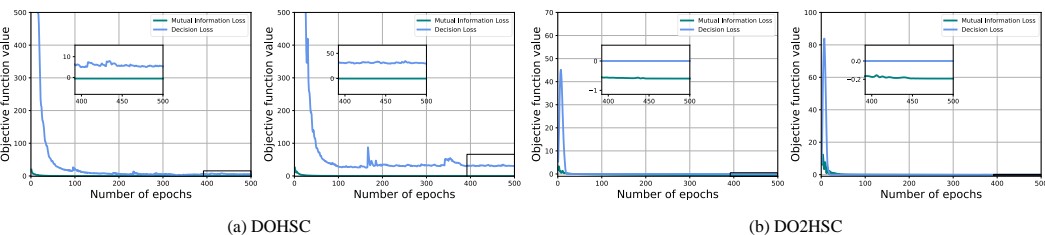

(a) DOHSC                   (b) DO2HSC

Figure 7: Convergence curves of the proposed models on the MUTAG dataset.

Regarding our model DOHSC, we firstly set 1 epoch in the pretraining stage to initialize the center of the decision boundary and then train the model in 500 epochs. The convergence curves are given in Figure 7 to indicate that the final optimized results are adopted. The improved method DO2HSC is also set 1-epoch pretraining stage and trains DOHSC 5 epochs to initialize a suitable center and decision boundaries $r_{max}$ and $r_{min}$, where the percentile $\nu$ of $r_{max}$ is fixed to $0.05$. After initializing, the model is trained in 500 epochs. For both proposed approaches, the trade-off factor $\lambda$ is set to 10 to ensure decision loss as the main optimization objective. Dimensions of the GIN hidden layer and orthogonal projection layer are fixed as 16 and 8, respectively. About the backbone network, a 4-layer GIN and a 3-layer fully connected neural network are adopted. Besides, the averages and standard deviations of the Area Under Operating Characteristic Curve (AUC) are used to support the comparable experiments by repeating each algorithm ten times. The higher value of the AUC metric represents better performance. When calculating the AUC of graph-kernel baselines, we estimated the radius of the hypersphere as 99 percentile of all squared distances to the separating hyperplane and then determined the score as the difference between squared distances and its square radius. Regarding our model DOHSC, we firstly set 1 epoch in the pretraining stage to initialize the center of the decision boundary and then train the model in 500 epochs. The convergence curves are given in Figure 7. The improved method DO2HSC is also set 1-epoch pretraining stage and trains DOHSC 5 epochs to initialize a suitable center and decision boundaries $r_{max}$ and $r_{min}$, where the percentile $\nu$ of $r_{max}$ is fixed to $0.1$. After initializing, the model is also trained in 500 epochs. For both proposed approaches, the trade-off factor $\lambda$ is set to 10. Dimensions of the GIN hidden layer and orthogonal projection layer are fixed as 16 and 8, respectively. About the backbone network, a 4-layer GIN and a 3-layer fully connected neural network are adopted. Besides, the averages and standard deviations of the Area Under Operating Characteristic Curve (AUC) are used to support the comparable experiments by repeating each algorithm ten times. The higher value of the AUC metric represents better performance. When calculating the AUC of graph-kernel baselines, we estimated the radius of the hypersphere as 99 percentile of all squared distances to the separating hyperplane and then determined the score as the difference between squared distances and its square radius.

## E  DOHSC AND DO2HSC ON NON-GRAPH DATA

Since our DOHSC and DO2HSC can also be applied to non-graph data such as images, here we compare them with some state-of-the-art anomaly detections Ruff et al. (2018); Goyal et al. (2020);

Liznerski et al. (2021) on Fashion-MNIST. The results are reported in Table 7. First, DOHSC and DO2HSC obtained seven best AUCs out of ten in total. And in the remaining three classes, the proposed models still achieved comparable performance with gaps of less than 2%. Second, Deep SVDD plays an important baseline role relative to DOHSC and DOHSC defeats it by a large margin in all classes. It further verifies that the proposed orthogonal projection is meaningful and helpful. In general, bi-hypersphere learning also performs sufficiently on common datasets and is very competitive compared to these state-of-the-art anomaly detection algorithms (Deep SVDD, DROCC and FCDD). In terms of the average AUC values for all classes of the dataset in Table 8, our algorithm outperforms all the compared baselines reproduced above, but it is worth mentioning that the reported performance of the IGD (Scratch) algorithm (Chen et al., 2022) is superior to our algorithm with no more than 2% gap.

Table 7: Average AUCs in one-class anomaly detection on Fashion-MNIST.

| Normal Class | Deep SVDD (Ruff et al., 2018) | DROCC (Goyal et al., 2020) | FCDD (Liznerski et al., 2021) | DOHSC | DO2HSC |
|---|---|---|---|---|---|
| T-shirt | $0.8263 \pm 0.0342$ | $0.8931 \pm 0.0072$ | $0.8143 \pm 0.0101$ | $0.9153 \pm 0.0082$ | $\mathbf{0.9196 \pm 0.0064}$ |
| Trouser | $0.9632 \pm 0.0072$ | $0.9835 \pm 0.0054$ | $\mathbf{0.9855 \pm 0.0014}$ | $0.9817 \pm 0.0060$ | $0.9839 \pm 0.0020$ |
| Pullover | $0.7885 \pm 0.0398$ | $0.8656 \pm 0.0140$ | $0.8423 \pm 0.0052$ | $0.8007 \pm 0.0204$ | $\mathbf{0.8768 \pm 0.0122}$ |
| Dress | $0.8607 \pm 0.0124$ | $0.8776 \pm 0.0269$ | $0.9143 \pm 0.0120$ | $\mathbf{0.9178 \pm 0.0230}$ | $0.9171 \pm 0.0084$ |
| Coat | $0.8417 \pm 0.0366$ | $0.8453 \pm 0.0143$ | $0.8607 \pm 0.0213$ | $0.8805 \pm 0.0258$ | $\mathbf{0.9038 \pm 0.0140}$ |
| Sandal | $0.8902 \pm 0.0281$ | $\mathbf{0.9336 \pm 0.0123}$ | $0.9089 \pm 0.0165$ | $0.8932 \pm 0.0287$ | $0.9308 \pm 0.0070$ |
| Shirt | $0.7507 \pm 0.0158$ | $0.7789 \pm 0.0188$ | $0.7750 \pm 0.0038$ | $\mathbf{0.8177 \pm 0.0124}$ | $0.8022 \pm 0.0045$ |
| Sneaker | $0.9676 \pm 0.0062$ | $0.9624 \pm 0.0059$ | $\mathbf{0.9874 \pm 0.0007}$ | $0.9678 \pm 0.0050$ | $0.9677 \pm 0.0075$ |
| Bag | $0.9039 \pm 0.0355$ | $0.7797 \pm 0.0749$ | $0.8584 \pm 0.0222$ | $\mathbf{0.9122 \pm 0.0258}$ | $0.9090 \pm 0.0105$ |
| Ankle Boot | $0.9488 \pm 0.0207$ | $0.9589 \pm 0.0207$ | $0.9432 \pm 0.0050$ | $0.9756 \pm 0.0127$ | $\mathbf{0.9785 \pm 0.0038}$ |

Table 8: Mean AUC of all classes on Fashion-MNIST.

| | Deep SVDD | DROCC | FCDD | IGD (Chen et al., 2022) | DOHSC | DO2HSC |
|---|---|---|---|---|---|---|
| Mean AUC | 0.8742 | 0.8879 | 0.8890 | **0.9201** | 0.9063 | 0.9189 |

## F SUPPLEMENTED VISUALIZATION

This part shows the related supplemented visualization results of the anomaly detection task.

From Section 4, we can see some DOHSC results are improved a lot through DO2HSC. For example, compared with DOHSC, DO2HSC often improves the results by less than 2% on most of the datasets. But on class 1 of ER_MD, DO2HSC has a more than 20% improvement. Here the distance distributions of DOHSC and DO2HSC on ER_MD are given in Figure 8 to prove this improvement. In Subfigure (a), the anomalous data appear in the distance interval [0,1], especially in the region close to the center, and less or even none of the normal data distributes in it. On the contrary, DO2HSC divided normal data and anomalous data more specifically, and both sides of the interval have anomalous data, as we assumed before.

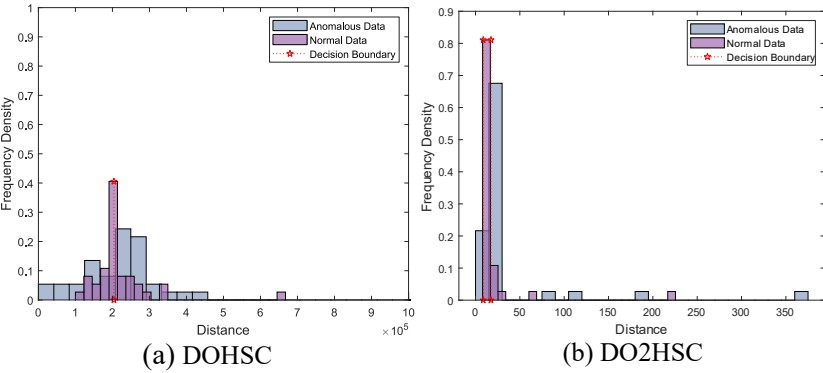

(a) DOHSC      (b) DO2HSC

Figure 8: Visualizations of DOHSC and DO2HSC on ER_MD dataset.

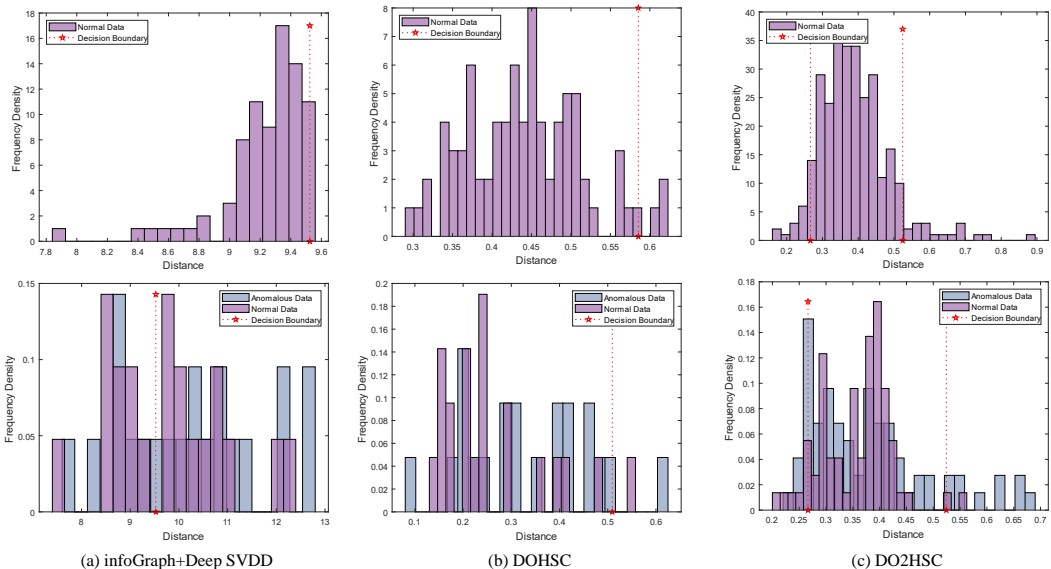

Figure 9: Distance distributions were obtained by infoGraph+Deep SVDD, the proposed model, and the improved proposed model on COX2. The first row represents the distance distribution of the training samples in relation to the decision boundary. The last row indicates the distance distribution of the test data with respect to the decision boundary.

Figure 9 shows the distance distributions of the two-stage method, the proposed model DOHSC, and the improved DO2HSC. Here, the *distance* is defined as the distance between the sample and the center of the decision hypersphere. The distance distribution denotes the sample proportion in this distance interval to the corresponding total samples. It can be intuitively observed that most distances of instances are close to the decision boundary because of the fixed learned representation. As mentioned earlier, the jointly-trained algorithm has mitigated the situation, and the obtained representation makes many instances have smaller distances from the center of the sphere.

However, like we wrote in Section 2, anomalous data may occur in regions with less training data, especially the region close to the center, which is also confirmed by (a) and (b) of Figure 9. Differently, DO2HSC effectively shrinks the decision area, and we find that the number of outliers is obviously reduced due to a more compact distribution of training data.

### F.1 PARAMETER SENSITIVITY AND ROBUSTNESS

To claim the stability of our models, we analyze the parameter sensitivity and robustness of DOHSC and DO2HSC, respectively. Consider that the projection dimension varies in {4, 8, 16, 32, 64, 128} while the hidden layer dimension of the GIN module ranges from 4 to 128. In Figure 11, the DO2HSC model has less volatile performances than DOHSC, especially when the training dataset is sampled from COX2 class 0, as Subfigure (d) shows. Noticeably, a higher dimension of the GIN hidden layer usually displays a better AUC result since the quality of learned graph representations improves when the embedding space is large enough.

In addition, we assess different aspects of model robustness. More specifically, the AUC results about two "ratios" are displayed: 1) Different sampling ratios for the training set; 2) Different ratios of noise disturbance for the learned representation. In Subfigures (c) (f), the purple bars regard normal data as class 0, while green bars treat normal data as class 1. Notice that most AUC results are elevating along with a higher ratio of authentic data in the training stage, demonstrating our models' potential in the unsupervised setting. On the other hand, when more noise is blended into the training dataset, the AUC performances of yellow line and blue line always stay stable at a high level. This outcome verifies our models' robustness in response to the alien data.

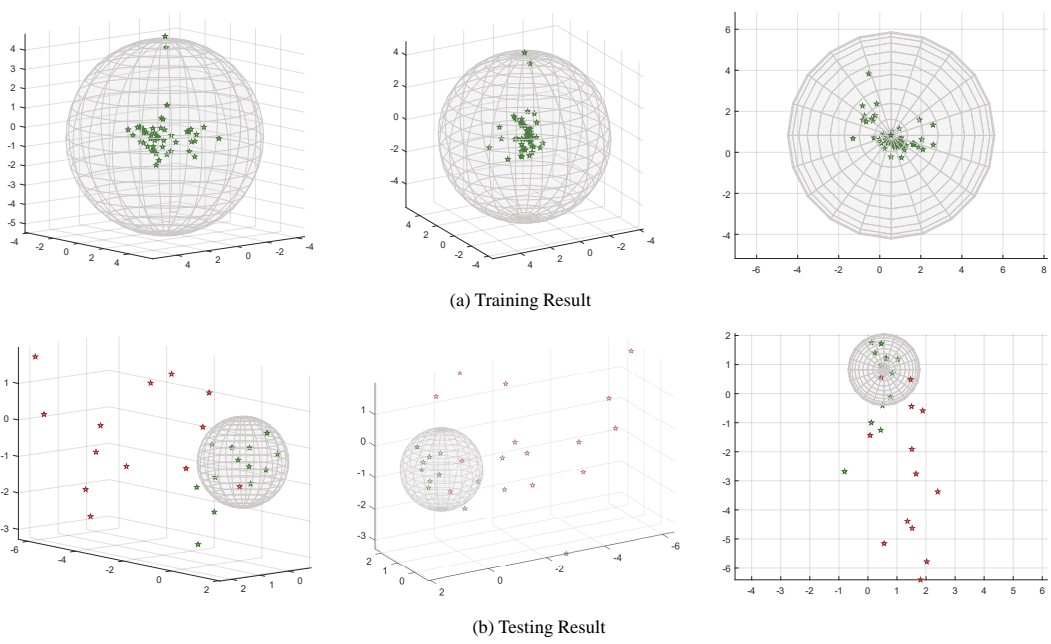

(a) Training Result

(b) Testing Result

Figure 10: Visualization results of the DOHSC with MUTAG in different perspectives.

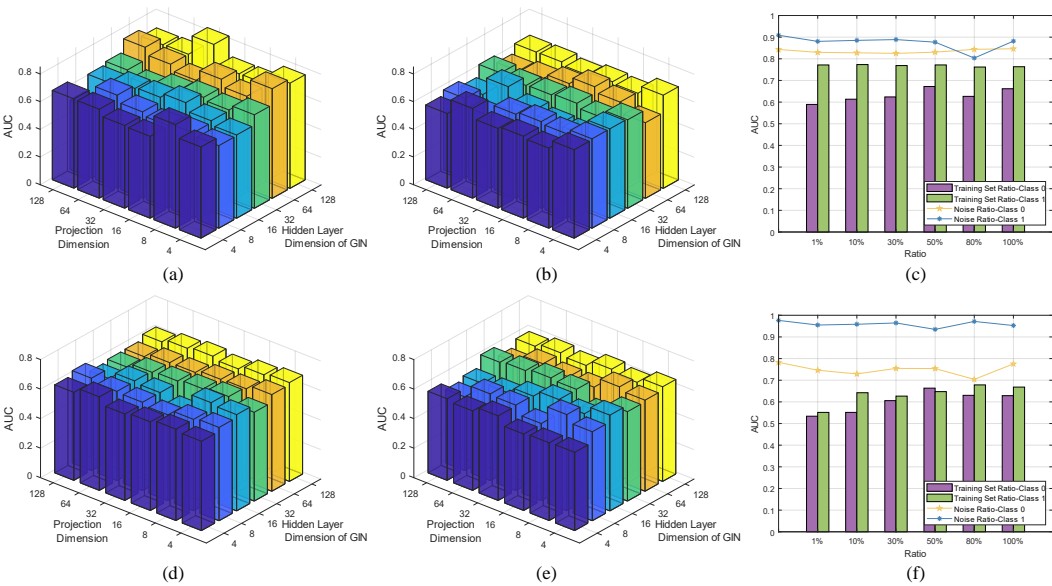

Figure 11: Parameter sensitivity and robustness of the proposed models. (a)-(b) Parameter sensitivity of DOHSC with different hidden layer dimensions of GIN and projection dimensions on COX2 with Class 0 and Class 1, respectively. (d)-(e) Parameter sensitivity of DO2HSC with the same settings. (c) and (f) shows the performance impacts with different ratios of the training set on the IMDB-Binary dataset and added noise disturbances on the MUTAG dataset while training DOHSC and DO2HSC, respectively.

The percentile parameter sensitivity is also given in this part. It is worth mentioning that we test DOHSC with varying percentile in $\{0.01, 0.1, ..., 0.8\}$ and test DO2HSC only in $\{0.01, 0.05, 0.1\}$ because two radii of DO2HSC is obtained by percentile $\nu$ and $1 - \nu$. Two radii will be equal when $\nu = 0.5$ and the interval between two co-centered hyperspheres will disappear. From the table, the performance would decrease when a larger percentile is set obviously. For example, on the MUTAG

dataset, setting the percentile as 0.01 is more beneficial for DOHSC than setting it as 0.8, and setting the percentile as 0.01 is better than setting it as 0.1 for DO2HSC due to the change of the interval area.

Table 9: Parameter sensitivity of the proposed methods with different percentiles (all normal data is set to Class 0.

| Method | Dataset/Percentile | 0.01 | 0.1 | 0.5 | 0.8 |
|---|---|---|---|---|---|
| DOHSC | MUTAG | **0.9527 (0.0187)** | 0.9497 (0.0249) | 0.9112 (0.0099) | 0.8675 (0.1287) |
| | COX2 | **0.6041 (0.0661)** | 0.6022 (0.0789) | 0.5232 (0.0494) | 0.5523 (0.0572) |
| Method | Dataset/Percentile | 0.01 | 0.05 | 0.1 | - |
| DO2HSC | MUTAG | **0.9089 (0.0609)** | 0.8041 (0.1006) | 0.6769 (0.1207) | – |
| | COX2 | **0.6329 (0.0292)** | 0.6149 (0.0187) | 0.5830 (0.0713) | – |

## G   SUPPLEMENTED RESULTS OF ABLATION STUDY

First, the ablation study of whether orthogonal projection needs standardization is conducted. To be more precise, we are pursuing orthogonal features, i.e., finding a projection matrix for orthogonal latent representation (with standardization) instead of computing the projection onto the column or row space of the projection matrix (non-standardization), though they are closely related to each other. This is equivalent to performing PCA and using the standardized principal components. Therefore, we show the comparison between DOHSC with standardization and without standardization. From Table 10, it is observed that the performance of DOHSC without standardization is acceptable and most results of it are better than the two-stage baseline, i.e., infoGraph+Deep SVDD. It verifies the superiority of the end-to-end method over the two-stage baselines. However, the model with standardization outperforms the non-standardized one in almost all cases.

Table 10: Comparison of the orthogonal projection layer with or w/o standardization.

| | Class | infoGraph+Deep SVDD | DOHSC (Non-Standardization) | DOHSC |
|---|---|---|---|---|
| MUTAG | 0 | $0.8805 \pm 0.0448$ | $0.8521 \pm 0.0650$ | $\mathbf{0.8822 \pm 0.0432}$ |
| | 1 | $0.6166 \pm 0.2052$ | $0.6918 \pm 0.1467$ | $\mathbf{0.8115 \pm 0.0279}$ |
| COX2 | 0 | $0.4825 \pm 0.0624$ | $0.5800 \pm 0.0473$ | $\mathbf{0.6263 \pm 0.0333}$ |
| | 1 | $0.5029 \pm 0.0700$ | $0.5029 \pm 0.0697$ | $\mathbf{0.6805 \pm 0.0168}$ |
| ER_MD | 0 | $0.5312 \pm 0.1545$ | $0.4881 \pm 0.0626$ | $\mathbf{0.6620 \pm 0.0308}$ |
| | 1 | $0.5682 \pm 0.0704$ | $\mathbf{0.6740 \pm 0.0356}$ | $0.5184 \pm 0.0793$ |
| DD | 0 | $0.3942 \pm 0.0436$ | $0.4029 \pm 0.0354$ | $\mathbf{0.7083 \pm 0.0188}$ |
| | 1 | $0.6484 \pm 0.0236$ | $0.6903 \pm 0.0215$ | $\mathbf{0.7579 \pm 0.0154}$ |
| IMDB-Binary | 0 | $0.6353 \pm 0.0277$ | $0.5149 \pm 0.0655$ | $\mathbf{0.6609 \pm 0.0033}$ |
| | 1 | $0.5836 \pm 0.0995$ | $0.6505 \pm 0.0585$ | $\mathbf{0.7705 \pm 0.0045}$ |
| COLLAB | 0 | $0.5662 \pm 0.0597$ | $0.6067 \pm 0.1007$ | $\mathbf{0.9185 \pm 0.0455}$ |
| | 1 | $0.7926 \pm 0.0986$ | $0.8958 \pm 0.0141$ | $\mathbf{0.9755 \pm 0.0030}$ |
| | 2 | $0.4062 \pm 0.0978$ | $0.4912 \pm 0.2000$ | $\mathbf{0.5450 \pm 0.0469}$ |

Besides, the ablation study of using the mutual information maximization loss is shown in Table 11. It can be intuitively concluded that mutual information loss does not always have a positive impact on all data. This also indicates that the anomaly detection optimization method and orthogonal projection we designed are effective instead of entirely due to the loss of mutual information.

To demonstrate the effectiveness of the orthogonal projection layer (OPL), we conduct ablation studies and visualize the comparison of 3-dimensional results produced with OPL and without OPL, respectively. For each model trained on a particular dataset class, we show the result without OPL on the left side, while the result with OPL is displayed on the right. As Figure 12 illustrates, OPL drastically improves the distribution of the embeddings to be more spherical rather than elliptical. Similarly, with the help of OPL, other embeddings show a more compact and rounded layout.

Table 11: Comparison of the loss supervision with or w/o **m**utual **i**nformation **l**oss (**MIL**).

|  | Class | DOHSC (Non-**MIL**) | DOHSC | DO2HSC (Non-**MIL**) | DO2HSC |
|---|---|---|---|---|---|
| MUTAG | 0 | **0.9456 ± 0.0189** | 0.8822 ± 0.0432 | 0.8308 ± 0.0548 | **0.9089 ± 0.0609** |
|  | 1 | 0.7597 ± 0.0802 | **0.8115 ±0.0279** | 0.7915 ± 0.0274 | **0.8250 ± 0.0790** |
| COX2 | 0 | **0.6349 ± 0.0466** | 0.6263 ± 0.0333 | 0.6143 ± 0.0302 | **0.6329 ± 0.0292** |
|  | 1 | 0.6231 ± 0.0501 | **0.6805 ± 0.0168** | **0.6576 ± 0.1830** | 0.6518 ± 0.0481 |
| ER_MD | 0 | 0.5837 ± 0.0778 | **0.6620 ± 0.0308** | 0.5836 ± 0.0909 | **0.6867 ± 0.0226** |
|  | 1 | **0.6465 ± 0.0600** | 0.5184 ± 0.0793 | **0.7424 ± 0.0385** | 0.7351 ± 0.0159 |
| DD | 0 | 0.4738 ± 0.0412 | **0.7083 ± 0.0188** | 0.6882 ± 0.0221 | **0.7320 ± 0.0194** |
|  | 1 | 0.7197 ± 0.0185 | **0.7579 ± 0.0154** | 0.7376 ± 0.0244 | **0.7651 ± 0.0317** |
| IMDB-Binary | 0 | 0.5666 ± 0.0810 | **0.6609 ± 0.0033** | 0.6303 ± 0.0538 | **0.6406 ± 0.0642** |
|  | 1 | 0.6827 ± 0.0239 | **0.7705 ± 0.0045** | 0.6810 ± 0.0276 | **0.7101 ± 0.0429** |
| COLLAB | 0 | **0.9330 ± 0.0539** | 0.9185 ± 0.0455 | 0.5415 ± 0.0182 | **0.6718 ± 0.0353** |
|  | 1 | 0.9744 ± 0.0017 | **0.9755 ± 0.0030** | **0.9293 ± 0.0023** | 0.9153 ± 0.0070 |
|  | 2 | **0.8275 ± 0.0765** | 0.5450 ± 0.0469 | **0.8452 ± 0.0243** | 0.7188 ± 0.0260 |

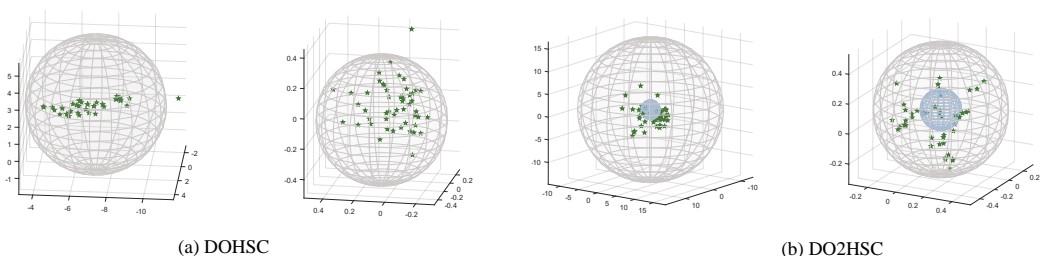

(a) DOHSC                     (b) DO2HSC

Figure 12: Visualizations on the MUTAG dataset Class 0 (left: with OPL; right: without OPL).

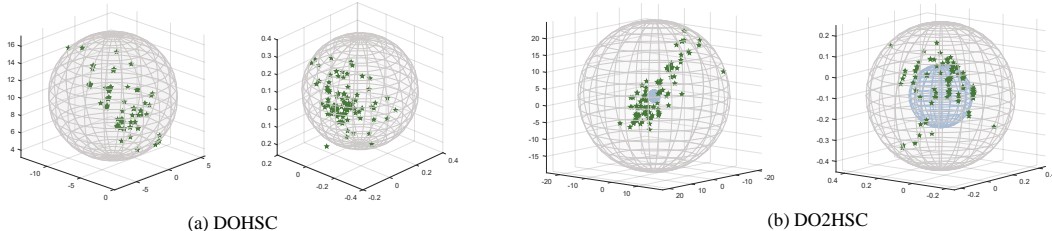

(a) DOHSC                     (b) DO2HSC

Figure 13: Visualizations on the MUTAG dataset Class 1 (left: with OPL; right: without OPL).

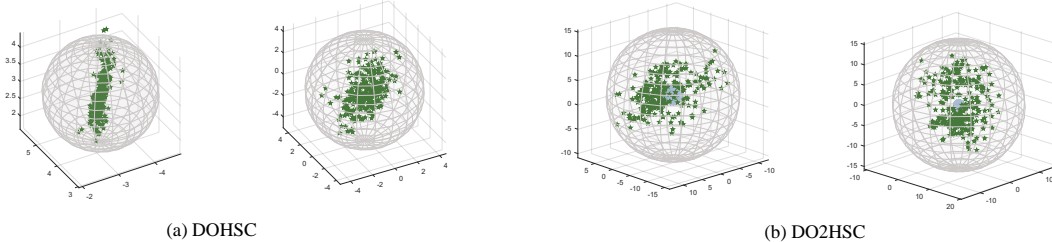

(a) DOHSC                     (b) DO2HSC

Figure 14: Visualizations on the COX2 dataset Class 0 (left: with OPL; right: without OPL).

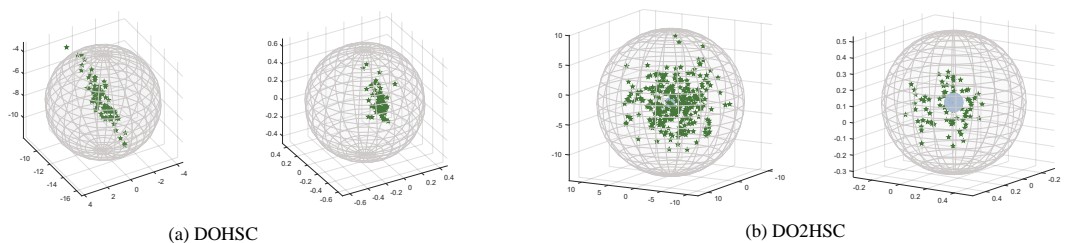

(a) DOHSC    (b) DO2HSC

Figure 15: Visualizations on the COX2 dataset Class 1 (left: with OPL; right: without OPL).

