# OpenReview forum: "Deep Graph-Level Orthogonal Hypersphere Compression for Anomaly Detection"
_ICLR.cc/2023/Conference — Submitted to ICLR 2023_

### Official Review · Reviewer_fADg · 2022-10-20

**Confidence:** 1
**Correctness:** 3
**Technical Novelty And Significance:** 3
**Empirical Novelty And Significance:** Not applicable
**Recommendation:** 6

**Clarity, Quality, Novelty And Reproducibility:**

I am unable to assess this paper and have alerted the ACs to seek an opinion from different reviewers.

**Strength And Weaknesses:**

I am unable to assess this paper and have alerted the ACs to seek an opinion from different reviewers.

**Summary Of The Paper:**

This paper first proposes a novel deep graph-level anomaly detection model, which learns the graph representation with maximum mutual information between substructure features and global structure features while exploring a hypersphere anomaly decision boundary. The numerical and visualization results on a few graph datasets demonstrate the effectiveness and superiority of the methods.

**Summary Of The Review:**

I am unable to assess this paper and have alerted the ACs to seek an opinion from different reviewers.

---

> ### Author Response · Authors · 2022-11-17
> **Thanks for your recognition.**
>
> We have carefully revised our paper. Welcome any suggestions or comments that you would like to provide!

---

### Official Review · Reviewer_bgGj · 2022-10-23

**Confidence:** 4
**Correctness:** 4
**Technical Novelty And Significance:** 3
**Empirical Novelty And Significance:** 3
**Recommendation:** 5

**Clarity, Quality, Novelty And Reproducibility:**

This paper is well-written and very clear. Since source codes are provided in the supp, it seems possible to reproduce the results.

**Strength And Weaknesses:**

Strength:
1.	The proposed method is simple yet effective. It seems possible to apply the proposed orthogonal projection layer and the bi-hypersphere compression model to many other graph anomaly detection methods.
2.	The paper is well-written and easy to follow. The main contributions are clearly summarized.
3.	Experiments on multiple datasets show the effectiveness of the proposed method.
Weaknesses:
1.	The motivation for the bi-hypersphere compression is still difficult to understand. The authors claim that anomalous data may appear in the empty inner decision region. But Figure 3 did not show this phenomenon. More importantly, the authors did not explain why bi-hypersphere compression can help. For example, if the normal area becomes more compact, how can we guarantee that anomalous data will not appear in the empty inner decision region? The authors may need to provide some comparisons to show the proposed method can indeed solve this problem.
2.	Improvements on some of the datasets seem unreasonable. For example, in Table3, compared with DOHSC, DO2HSC often improves the results by less than 2% on most of the datasets. But on class 1 of ER_MD, DO2HSC has a more than 20% improvement. Analysis needs to be provided. For example, it will be very interesting to show, on ER_MD, more anomalous data will appear in the empty inner decision region so that DO2HSC has an impressive improvement on this dataset.


**Summary Of The Paper:**

This paper proposed a graph anomaly detection method through mutual information maximization. The key contributions are proposing (i) an orthogonal projection layer for the decision boundary correction and (ii) a two co-centered hyperspheres structure for estimating the normal distribution. Experimental results on multiple datasets show the effectiveness of the proposed method.

**Summary Of The Review:**

This paper proposed a simple yet effective method for graph anomaly detection. Authors may consider the weakness shown above and revise the paper.

---

> ### Author Response · Authors · 2022-11-17
> **Thanks for the review. We analyze our motivation for bi-hypersphere compression and show the corresponding comparisons with others.**
>
> **Response to questions:**
>
> **Q1: The motivation for the bi-hypersphere compression is still difficult to understand. The authors claim that anomalous data may appear in the empty inner decision region. But Figure 3 did not show this phenomenon. More importantly, the authors did not explain why bi-hypersphere compression can help. For example, if the normal area becomes more compact, how can we guarantee that anomalous data will not appear in the empty inner decision region? The authors may need to provide some comparisons to show the proposed method can indeed solve this problem.**
>
> **Response:** Thank you for your careful comments and bringing up the discussion about the intuition behind bi-hypersphere compression.
>
> First, from Figure 3, we do see that anomalous data fall in areas with few or even no distribution of training data, such as interval [0,0.2]. It commonly happens to the region close to the centroid. In addition, most training data are often far from the centroid. Thus, we want to reduce the volume of the space enclosing the training data.
>
> With regard to bi-hypersphere learning how to help that phenomenon, the empirical findings are actually related to the characteristics of high-dimensional distributions. The anomaly score is determined by the $\ell_2$ norm $\Vert\mathbf{z}-\mathbf{c}\Vert$, where $\mathbf{c}$ denotes the centroid. Let $\tau$ be a threshold determined by a certain quantile. If $\Vert\mathbf{z}-\mathbf{c}\Vert\geq \tau$, then $\mathbf{z}$ is abnormal; otherwise, $\mathbf{z}$ is normal. However, most data points satisfy $\tau'\leq \Vert\mathbf{z}-\mathbf{c}\Vert$, where $\tau'$ is close to $\tau$ especially when the dimension is high. This indicates that within the range $[0,\tau']$, the normality is not supported by the training data. Take multivariate Gaussian simulation (Please see Figure 6 in Appendix B) as an example. Since the peak (mode) of the probability density function is at the origin, it is natural to expect most samples to be near the origin. However, in the high dimensions, the typical set (where data has information closest to the expected entropy of the population) of a Gaussian is the thin shell within a distance from the origin. To be more precise, we show the following table, in which we consider multivariate Gaussians with different dimensions. We see that the higher the dimension is, the greater the quantities of data further away from the center.
> Actually for any $d$-dimensional Gaussian $\mathcal{N}(\mathbf{0},\mathbf{I}_d)$, (using Lemma 1 of Laurent and Massart 2000), we can prove that
> $\mathbb{P}\left[\Vert \mathbf{z}\Vert \geq \sqrt{d-2\sqrt{dt}}\right] \geq 1-e^{-t}$,  for all $t > 0$, which matches the values in the table.
>
> \begin{matrix}
> \hline
>  Quantile \\;(correspond\\;to\\;\tau' or \ \tau) & dim=1  & dim=10 & dim=50 & dim=100 & dim=200 & dim=500 \\\\ \hline
> 0.01 & 0.0127 & 1.5957 & 5.5035 & 8.3817  & 12.5117 & 20.6978 \\\\
> 0.25 & 0.3115 & 2.5829 & 6.5380  & 9.4908  & 13.6247 & 21.8542 \\\\
> 0.50 & 0.6671 & 3.0504 & 7.0141 & 9.9662  & 14.1054 & 22.3337 \\\\
> 0.75 & 1.1471 & 3.5399 & 7.5032 & 10.4386 & 14.5949 & 22.8200   \\\\
> 0.99 & 2.5921 & 4.8265 & 8.7723 & 11.6049 & 15.7913 & 24.0245 \\\\\hline
> \end{matrix}
> As shown in the table, when the dimension is higher, $\tau'$ is closer to $\tau$, namely, more data are far from the centroid.
> The related explanations and numerical simulations can be referred to Appendix B of our paper. We also give two references to support our thoughts as follows.
>
> **References:**
>
> [1] Roman Vershynin. 2018. Similarity of normal and spherical distributions. In High-Dimensional Probability An Introduction with Applications in Data Science. Cambridge University Press, 53-53.
>
> [2] Mianzhi Wang. 2018. The Counterintuitive Behavior of High-Dimensional Gaussian. research.wmz.ninja. https://research.wmz.ninja/articles/2018/03/the-counterintuitive-behavior-of-high-dimensional-gaussian-distributions.html
>
> We also give a comparison in Figure 9, Appendix F, that compared to infoGraph+Deep SVDD and DOHSC, DO2HSC can avoid this problem to some extent indeed. Taking Subfigure (b) as an example, fewer training samples distribute on the distance vary in $[0,0.31]$ at the first row, while some anomalous data (second row) appears in this range obviously.
> On the contrary, DO2HSC obtains a better result. From the training data distribution, the sample distribution is more concentrated, and most data are distributed within the interval.

---

> > ### Author Response · Authors · 2022-11-17
> > **Thanks for the review. We added the visualization result of ER_MD dataset.**
> >
> > **Q2: Improvements on some of the datasets seem unreasonable. For example, in Table 3, compared with DOHSC, DO2HSC often improves the results by less than 2\% on most of the datasets. But on class 1 of ER\_MD, DO2HSC has a more than 20\% improvement. Analysis needs to be provided. For example, it will be very interesting to show, on ER\_MD, more anomalous data will appear in the empty inner decision region so that DO2HSC has an impressive improvement on this dataset.**
> >
> > **Response:** Thank you for your instructive suggestions. We have visualized the distance distributions of ER$\\_$MD, obtained by DOHSC and DO2HSC methods in Figure 8, Appendix F. In Subfigure (a), the anomalous data appear in the distance interval [0,1], especially in the region close to the center. On the contrary, DO2HSC divided normal data and anomalous data more precisely, and both sides of the interval have anomalous data, as we assumed before.
> >
> > **Thank you again for recognizing our work.**

---

### Official Review · Reviewer_P4g5 · 2022-10-25

**Confidence:** 5
**Correctness:** 2
**Technical Novelty And Significance:** 2
**Empirical Novelty And Significance:** Not applicable
**Recommendation:** 3

**Clarity, Quality, Novelty And Reproducibility:**

The overall paper clarity is fairly good. One of the components is new, but the advantages of this new one-class learning design compared to recently proposed ones are unclear.

**Strength And Weaknesses:**

Strengths of the paper are:
- The work tackles an important yet under-explored problem -- graph-level anomaly detection. Unlike node anomaly detection, graph-level anomaly detection methods, especially deep neural network-based methods, are relatively limited.
- The bi-hypersphere learning objective is an improved version of popular deep one-class classifiers, which is new, to the best of my knowledge.
- The effectiveness of the SVD-based representation projection and the bi-hypersphere learning is justified via the ablation study.

The negative aspects of the paper include:
- Although the paper is focused on graph-level anomaly detection, the key design or the newly proposed component (i.e., the bi-hypersphere learning) is generic and does not take into account of the graph-level graph mining tasks. The only component relevant to graph-level detection is the mutual information maximization between local and global representations, which is taken directly from existing work, such as Infograph. The main concern here is that the presented method is not designed specifically for graph-level anomaly detection.
- The bi-hypersphere learning seems to be generalizable to different types of data. Results on image/tabular data would be important  to justify whether the main argument of the bi-hypersphere learning is effective.
- There have been many variants of deep one-class classifier for learning more meaningful one-class models, such as [1-4]. They can be easily combined with loss functions like Infograph to adapt to graph-level anomaly detection. It's unclear what are the advantages of the proposed method compared to these more advanced one-class classification methods.
- The datasets used are very different from the ones in the competing methods like GLocalKD and OCGTL. Their performance seems to be less effective than the ones reported in their original paper. What are the reasons/motivations behind? GLocalKD and OCGTL seem to work well on some commonly used graph datasets, e.g., PROTEINS, AIDS, and REDDIT. How is the performance of the proposed method on those datasets?
- The argument that "In contrast, there is little work on graph data despite the fact ..." is invalid. To my knowledge, there have been many studies on anomalous node detection; less work is on graph-level anomaly detection.
- The benefit of using the mutual information maximization loss is not examined
- To my understanding, the method can be sensitive to the setting of the percentile parameter in eq. 14, but no empirical results are given about this sensitivity.

**References**
- [1] "DROCC: Deep robust one-class classification." In International Conference on Machine Learning, pp. 3711-3721. PMLR, 2020.
- [2] "Learning and evaluating representations for deep one-class classification." arXiv preprint arXiv:2011.02578 (2020).
- [3]  "Explainable deep one-class classification." arXiv preprint arXiv:2007.01760 (2020).
- [4] "Deep one-class classification via interpolated gaussian descriptor." In Proceedings of the AAAI Conference on Artificial Intelligence, vol. 36, no. 1, pp. 383-392. 2022.

**Summary Of The Paper:**

The work presents a new one-class classification method for graph-level anomaly detection. The new components added to the original one-class objective include a local-global graph representation learning, a SVD-based representation projection, and a bi-hypersphere learning. The first two newly added components are directly taken from existing work. The bi-hypersphere learning is new, as far as I know. The method is evaluated on six graph datasets and compared with graph kernel and GNN-based methods.

**Summary Of The Review:**

The paper has some merits, but the cons outweigh the pros, as discussed above.

---

> ### Author Response · Authors · 2022-11-17
> **Thanks for the review. We added the experimental results on Fashion-MNIST dataset, which show the superiority of our methods over the baselines on non-graph data.**
>
> **Response to questions:**
>
> **Q1: Although the paper is focused on graph-level anomaly detection, the key design or the newly proposed component (i.e., the bi-hypersphere learning) is generic and does not take into account of the graph-level graph mining tasks. The only component relevant to graph-level detection is the mutual information maximization between local and global representations, which is taken directly from existing work, such as Infograph. The main concern here is that the presented method is not designed specifically for graph-level anomaly detection.**
>
> **Response:** Thanks for your comments.
> First of all, existing graph-level anomaly detection methods are very few.
> Second, we believe that "the presented method is not designed specifically for graph-level anomaly detection" is not a shortcoming of our work. Besides graph data, our methods work well on other types of data.
>
> In our response to your next question, it can be found that our methods are at least as good as the state-of-the-art on non-graph data anomaly detection. This further verifies the significance of our work.
>
> **Q2: The bi-hypersphere learning seems to be generalizable to different types of data. Results on image/tabular data would be important to justify whether the main argument of the bi-hypersphere learning is effective.**
>
> **Response:** We tested the bi-hypersphere learning on the Fashion-MNIST dataset in the following table. The overall performance of bi-hypersphere learning is acceptable and competitive. For more details, please see our response to question 3.
>
> \begin{matrix}
> \hline
>     Normal\\,Class      & Deep SVDD           & DROCC\\,[1]    & FCDD\\,[2]                & DOHSC               & DO2HSC                               \\\\\hline
> T\text{-}shirt    & 0.8263 \pm 0.0342 & 0.8931 \pm 0.0072 & 0.8143 \pm 0.0101  & 0.9153
> \pm 0.0082 & \textbf{0.9196 $\pm$ 0.0064}                        \\\\
> Trouser    & 0.9632 \pm 0.0072 & 0.9835 \pm 0.0054 & \textbf{0.9855 $\pm$ 0.0014}  & 0.9817 \pm 0.0060 & 0.9839 \pm 0.0020                        \\\\
> Pullover   & 0.7885 \pm 0.0398 & 0.8656 \pm 0.0140 & 0.8423 \pm 0.0052  & 0.8007 \pm 0.0204 & \textbf{0.8768 $\pm$ 0.0122}                        \\\\
> Dress      & 0.8607 \pm 0.0124 & 0.8776 \pm 0.0269 & 0.9143 \pm 0.0120  & \textbf{0.9178 $\pm$ 0.0230} & 0.9171 \pm 0.0084 \\\\
> Coat       & 0.8417 \pm 0.0366 & 0.8453 \pm 0.0143 & 0.8607 \pm 0.0213  & 0.8805 \pm 0.0258 & \textbf{0.9038 $\pm$ 0.0140}                        \\\\
> Sandal     & 0.8902 \pm 0.0281 & \textbf{0.9336 $\pm$ 0.0123}& 0.9089 \pm 0.0165  & 0.8932 \pm 0.0287 & 0.9308 \pm 0.0070                        \\\\
> Shirt      & 0.7507 \pm 0.0158 & 0.7789 \pm 0.0188 & 0.7750 \pm 0.0038  & \textbf{0.8177 $\pm$ 0.0124} & 0.8022 \pm 0.0045                  \\\\
> Sneaker    & 0.9676 \pm 0.0062 & 0.9624 \pm 0.0059 & \textbf{0.9874 $\pm$ 0.0007}  & 0.9678 \pm 0.0050 & 0.9677 \pm 0.0075                        \\\\
> Bag        & 0.9039 \pm 0.0355 & 0.7797 \pm 0.0749 & 0.8584 \pm 0.0222  & \textbf{0.9122 $\pm$ 0.0258}  & 0.9090 \pm 0.0105                        \\\\
> Ankle\\,Boot & 0.9488 \pm 0.0207 & 0.9589 \pm 0.0207 & 0.9432 \pm 0.0050 & 0.9756 \pm 0.0127 & \textbf{0.9785 $\pm$ 0.0038}   \\\\\hline
>     \end{matrix}
>
> \begin{matrix}
> \hline
>     & Deep\\;SVDD\\,[4] & DROCC\\,[1] & FCDD\\,[2]& IGD\\,[3] & DOHSC & DO2HSC \\\\\hline
> Mean\\;AUC & 0.8742& 0.8879& 0.8890& \textbf{0.9201}    & 0.9063                           & 0.9189  \\\\\hline
> \end{matrix}
>
> **References:**
>
> [1] Sachin Goyal, Aditi Raghunathan, Moksh Jain, Harsha Vardhan Simhadri, and Prateek Jain. DROCC: deep robust one-class classification. In Proceedings of the 37th International Conference on Machine Learning, pp. 3711–3721, 2020.
>
> [2] Philipp Liznerski, Lukas Ruff, Robert A. Vandermeulen, Billy Joe Franks, Marius Kloft, and Klaus-Robert Muller. Explainable deep one-class classification. In Proceedings of the 9th International Conference on Learning Representations, 2021
>
> [3] Yuanhong Chen, Yu Tian, Guansong Pang, and Gustavo Carneiro. Deep one-class classification via
> interpolated gaussian descriptor. In Proceedings of the AAAI Conference on Artificial Intelligence,
> volume 36, pp. 383–392, 2022.
>
> [4] Lukas Ruff, Robert Vandermeulen, Nico Goernitz, Lucas Deecke, Shoaib Ahmed Siddiqui, Alexander Binder, Emmanuel Muller, and Marius Kloft. Deep one-class classification. In International Conference on Machine Learning, pp. 4393–4402, 2018.
>
> **It is worth mentioning that the idea of our bi-hypersphere learning is parallel to the ideas of [1][2][3][4]. Compared to Deep SVDD, the improvement given by our DO2HSC  is remarkable.**

---

> > ### Author Response · Authors · 2022-11-17
> > **Thanks for the review. We compare with several SOTA one-class classification methods.**
> >
> > **Q3: There have been many variants of deep one-class classifier for learning more meaningful one-class models, such as [1-4]. They can be easily combined with loss functions like Infograph to adapt to graph-level anomaly detection. It's unclear what are the advantages of the proposed method compared to these more advanced one-class classification methods.**
> >
> > [1] "DROCC: Deep robust one-class classification." In International Conference on Machine Learning, pp. 3711-3721. PMLR, 2020.
> >
> > [2] "Learning and evaluating representations for deep one-class classification." arXiv preprint arXiv:2011.02578 (2020).
> >
> > [3] "Explainable deep one-class classification." arXiv preprint arXiv:2007.01760 (2020).
> >
> > [4] "Deep one-class classification via interpolated gaussian descriptor." In Proceedings of the AAAI Conference on Artificial Intelligence, vol. 36, no. 1, pp. 383-392. 2022.
> >
> > **Response:** Here we reproduced the results of references [1], [3], and compared them with the reported average performance in [4]. As shown by the table, the proposed DOHSC and DO2HSC basically obtained the best results in seven classes. Even if performance is not optimal in the other three classes, the gaps from the best results are within acceptable limits. It is noted that Deep SVDD plays an important baseline role relative to DOHSC, and DOHSC defeats it by a large margin in all classes. It further proves that the proposed orthogonal projection is meaningful and helpful.
> > In general, bi-hypersphere learning also performs sufficiently on common datasets and is very competitive compared to these state-of-the-art anomaly detection algorithms (DROCC and FCDD).
> >
> > \begin{matrix}
> > \hline
> > Normal\\,Class      & Deep\\,SVDD           & DROCC\\,[1]    & FCDD\\,[3]                & DOHSC               & DO2HSC                               \\\\\hline
> > T\text{-}shirt    & 0.8263 \pm 0.0342 & 0.8931 \pm 0.0072 & 0.8143 \pm 0.0101  & 0.9153
> > \pm 0.0082 & \textbf{0.9196 $\pm$ 0.0064}                        \\\\
> > Trouser    & 0.9632 \pm 0.0072 & 0.9835 \pm 0.0054 & \textbf{0.9855 $\pm$ 0.0014}  & 0.9817 \pm 0.0060 & 0.9839 \pm 0.0020                        \\\\
> > Pullover   & 0.7885 \pm 0.0398 & 0.8656 \pm 0.0140 & 0.8423 \pm 0.0052  & 0.8007 \pm 0.0204 & \textbf{0.8768 $\pm$ 0.0122}                        \\\\
> > Dress      & 0.8607 \pm 0.0124 & 0.8776 \pm 0.0269 & 0.9143 \pm 0.0120  & \textbf{0.9178 $\pm$ 0.0230} & 0.9171 \pm 0.0084 \\\\
> > Coat       & 0.8417 \pm 0.0366 & 0.8453 \pm 0.0143 & 0.8607 \pm 0.0213  & 0.8805 \pm 0.0258 & \textbf{0.9038 $\pm$ 0.0140}                        \\\\
> > Sandal     & 0.8902 \pm 0.0281 & \textbf{0.9336 $\pm$ 0.0123}& 0.9089 \pm 0.0165  & 0.8932 \pm 0.0287 & 0.9308 \pm 0.0070                        \\\\
> > Shirt      & 0.7507 \pm 0.0158 & 0.7789 \pm 0.0188 & 0.7750 \pm 0.0038  & \textbf{0.8177 $\pm$ 0.0124} & 0.8022 \pm 0.0045                  \\\\
> > Sneaker    & 0.9676 \pm 0.0062 & 0.9624 \pm 0.0059 & \textbf{0.9874 $\pm$ 0.0007}  & 0.9678 \pm 0.0050 & 0.9677 \pm 0.0075                        \\\\
> > Bag        & 0.9039 \pm 0.0355 & 0.7797 \pm 0.0749 & 0.8584 \pm 0.0222  & \textbf{0.9122 $\pm$ 0.0258}  & 0.9090 \pm 0.0105                        \\\\
> > Ankle\\,Boot & 0.9488 \pm 0.0207 & 0.9589 \pm 0.0207 & 0.9432 \pm 0.0050 & 0.9756 \pm 0.0127 & \textbf{0.9785 $\pm$ 0.0038}   \\\\\hline
> >     \end{matrix}
> >
> > According to the results in the table, our methods cannot obtain better results than IGD [4], which is indeed a superior method. However, we want to claim that we are not aiming to look for SOTA results, we just attempt to make some improvements and innovations in this area. In fact, our proposed orthogonal projection and bi-hypersphere learning do achieve better results compared with the original hypersphere anomaly detection.
> >
> > \begin{matrix}
> > \hline
> >     & Deep\\;SVDD & DROCC\\,[1] & FCDD\\,[3]& IGD\\,[4] & DOHSC & DO2HSC \\\\\hline
> > Mean\\;AUC & 0.8742& 0.8879& 0.8890& \textbf{0.9201}    & 0.9063                           & 0.9189  \\\\\hline
> > \end{matrix}
> >
> > We have included the four references and the two tables in our paper (Appendix E).
> >
> > Moreover, we argue that our primary contribution is that DO2HSC method can alleviate the *soap-bubble* problem existing in outlier detection, which we illustrate in Appendix B of the paper. Please refer to our paper or our response to the first reviewer for a more detailed explanation.

---

> > > ### Author Response · Authors · 2022-11-17
> > > **Thanks for the review. We add and analyze the results of two commonly used graph datasets.**
> > >
> > > **Q4: The datasets used are very different from the ones in the competing methods like GLocalKD and OCGTL. Their performance seems to be less effective than the ones reported in their original paper. What are the reasons/motivations behind? GLocalKD and OCGTL seem to work well on some commonly used graph datasets, e.g., PROTEINS, AIDS, and REDDIT. How is the performance of the proposed method on those datasets?**
> > >
> > > **Response:** Thank you for your comments.
> > > For GLocalKD and OCGTL, we adopted their default experimental settings. Note that the dataset is divided differently, while we take eighty percent of each class as normal data for training, and the rest of the data with other categories of data in a 1:1 ratio to form the test data.
> > > GLocalKD reported their performance through treating the minority class as anomalies in their paper.
> > > Besides, due to the limited device, the relatively large-scale dataset is tested with a small batch size, such as DD. This may also lead to worse performance.
> > > But except for the aforementioned situations, the reproduced results are basically consistent with those reported values.
> > > Actually, we think it is a common phenomenon because sometimes suitable settings are necessary for an excellent performance.
> > >
> > > Besides, we provided the AUC results of AIDS and PROTEINS. REDDIT dataset brings an 'out-of-memory' error on our limited devices, so we are very sorry that we cannot give its performance. The experimental results are shown in the following table. From it we can see although the proposed two methods do not have the best results on AIDS and Class 0 of PROTEINS, the gaps do not exceed to 2\% on Class 1 of AIDS and Class 0 of PROTEINS. And we also obtain the best performance on PROTEINS Class 1. Therefore, it can be concluded that our performance is at least acceptable on these additional graph datasets.
> > >
> > > \begin{matrix}
> > > \hline
> > > & AIDS &   & PROTEINS &  \\\\ \hline
> > > & 0             & 1               & 0               & 1                       \\\\ \hline
> > > SP\+OCSVM & 0.9778 (0.0000)    & 0.2832 (0.0000)   & 0.6683 (0.0000)  &  0.5202 (0.0000)  \\\\
> > > WL\_2\+OCSVM                 & 0.8975 (0.0000) & 0.2341 (0.0000) & 0.7239 (0.0000)   & 0.4833 (0.0000)      \\\\
> > > WL\_5\+OCSVM  & 0.9319 (0.0000)& 0.1598 (0.0000)                      & \textbf{0.7319} (0.0000)  & 0.5007 (0.0000) \\\\
> > > WL\_8\+OCSVM  & 0.9381 (0.0000) & 0.1237 (0.0000)                   & 0.7121 (0.0000)  & 0.5019 (0.0000)       \\\\
> > > WL\_{10}\+OCSVM & 0.9378 (0.0000) & 0.1085 (0.0000)      & 0.7034 (0.0000)    &  0.4962 (0.0000)  \\\\
> > > NH\+OCSVM & 0.9685 (0.0021) & 0.4992 (0.0054)  &  0.6828 (0.0000) & 0.5551 (0.0000) \\\\
> > > RW\+OCSVM & 0.6503 (0.0312) & 0.4084 (0.0310)                   & -                     & -            \\\\ \hline
> > > OCGIN & 0.9065 (0.0204)        & 0.8152 (0.0376)               & 0.5501 (0.0965)                             & 0.4777 (0.0764)             \\\\
> > > infoGraph\+Deep\\;SVDD                  & 0.8417 (0.0550)          & 0.8741 (0.0227)                             & 0.6504 (0.1335) & 0.4702 (0.0692)                               \\\\
> > > GLocalKD                            & \textbf{0.9915 (0.0003)}                                             &  0.1742 (0.2109)        & 0.7212 (0.0008)  & 0.7480 (0.0012)                 \\\\
> > > OCGTL  & 0.9809 (0.0048)                                             & \textbf{0.9934 (0.0006)}   & 0.6320 (0.0540)  & 0.5810 (0.0610)\\\\ \hline
> > > DOHSC\\,(Ours)         & 0.8211 (0.1505)     & 0.9914 (0.0054)                                         & 0.5685 (0.0413)& \textbf{0.7624 (0.0138)}                      \\\\
> > > DO2HSC\\,(Ours)        & 0.9213 (0.0542)       &  0.9376 (0.0087)      & 0.7197 (0.0377)   &  0.7421 (0.0209)  \\\\ \hline
> > > \end{matrix}

---

> > > > ### Author Response · Authors · 2022-11-17
> > > > **Thanks for the review. We add the ablation study of mutual information loss and parameter sensitivity of percentile.**
> > > >
> > > > **Q5: The argument that "In contrast, there is little work on graph data despite the fact ..." is invalid. To my knowledge, there have been many studies on anomalous node detection; less work is on graph-level anomaly detection.
> > > > The benefit of using the mutual information maximization loss is not examined.**
> > > >
> > > > **Response:** Thank you for your careful comments. First, our real intention is to illustrate there is little work on graph-level anomaly detection. We will revise our statement carefully. Besides, as you suggested, the ablation study of using the mutual information maximization loss is shown as follows. It can be intuitively concluded that mutual information loss does not always have a positive impact on all data. This also indicates that the anomaly detection optimization method and orthogonal projection we designed are effective instead of entirely due to the loss of mutual information.
> > > >
> > > > \begin{matrix}
> > > > \hline
> > > >           & Class & DOHSC\\;(Non\text{-}\textbf{MIL}) & DOHSC  & DO2HSC\\;(Non\text{-}\textbf{MIL}) & DO2HSC                 \\\\\hline
> > > > MUTAG      & 0   & \textbf{0.9456 $\pm$ 0.0189} &  0.8822 \pm 0.0432 & 0.8308 \pm  0.0548    & \textbf{0.9089 $\pm$ 0.0609}   \\\\
> > > >                              & 1    & 0.7597 \pm 0.0802 &  \textbf{0.8115 $\pm$0.0279} & 0.7915  \pm  0.0274    & \textbf{0.8250  $\pm$ 0.0790}  \\\\\hline
> > > > COX2        & 0   & \textbf{0.6349 $\pm$ 0.0466} &  0.6263 \pm 0.0333  &  0.6143 \pm  0.0302      &  \textbf{0.6329 $\pm$ 0.0292}  \\\\
> > > >                              & 1   & 0.6231 \pm 0.0501 &  \textbf{0.6805 $\pm$ 0.0168}  & \textbf{0.6576 $\pm$ 0.1830} &  0.6518 \pm  0.0481                     \\\\\hline
> > > > ER\\_MD      & 0    &  0.5837 \pm 0.0778 &  \textbf{0.6620 $\pm$ 0.0308} &  0.5836 \pm  0.0909    & \textbf{0.6867 $\pm$ 0.0226}  \\\\
> > > >                              & 1   & \textbf{0.6465 $\pm$ 0.0600} &  0.5184 \pm 0.0793 & \textbf{0.7424 $\pm$  0.0385} & 0.7351 \pm 0.0159                         \\\\\hline
> > > > DD          & 0    & 0.4738 \pm 0.0412  & \textbf{0.7083  $\pm$ 0.0188} & 0.6882 \pm  0.0221   & \textbf{0.7320  $\pm$ 0.0194}  \\\\
> > > >                              & 1    & 0.7197 \pm 0.0185 &   \textbf{0.7579 $\pm$ 0.0154} &  0.7376 \pm 0.0244 &  \textbf{0.7651 $\pm$  0.0317}                    \\\\ \hline
> > > > IMDB\text{-}Binary & 0   & 0.5666 \pm 0.0810 &  \textbf{0.6609 $\pm$ 0.0033}  & 0.6303  \pm  0.0538   &  \textbf{0.6406
> > > >  $\pm$ 0.0642} \\\\
> > > >                              & 1   & 0.6827 \pm 0.0239 &  \textbf{0.7705 $\pm$  0.0045} & 0.6810 \pm   0.0276                      & \textbf{0.7101 $\pm$  0.0429} \\\\ \hline
> > > > COLLAB      & 0   & \textbf{0.9330 $\pm$ 0.0539} &  0.9185 \pm 0.0455  &  0.5415  \pm 0.0182 & \textbf{0.6718  $\pm$ 0.0353}  \\\\
> > > >                              & 1   & 0.9744 \pm 0.0017 &  \textbf{0.9755 $\pm$ 0.0030} &                \textbf{0.9293  $\pm$  0.0023}  &   0.9153  \pm  0.0070 \\\\
> > > >                              & 2    & \textbf{0.8275 $\pm$ 0.0765} &  0.5450  \pm 0.0469  &  \textbf{0.8452  $\pm$  0.0243}                      &  0.7188  \pm 0.0260 \\\\ \hline
> > > > \end{matrix}
> > > >
> > > > **Q6: To my understanding, the method can be sensitive to the setting of the percentile parameter in Eq. 14, but no empirical results are given about this sensitivity.**
> > > >
> > > > **Response:** Thank you for your suggestion. The ablation study of different percentiles is given. It is worth mentioning that we test DOHSC with varying percentile in $\\{0.01,0.1,...,0.8\\}$ and test DO2HSC only in $\\{0.01,0.05,0.1\\}$ because two radii of DO2HSC is obtained by percentile $\nu$ and $1-\nu$. Two radii will be equal when $\nu=0.5$ and the interval between two co-centered hyperspheres will disappear.
> > > > From the table, the performance would decrease when a larger percentile is set, obviously. For example, on the MUTAG dataset, setting the percentile as 0.01 is more beneficial for DOHSC than setting it as 0.8, and setting the percentile as 0.01 is better than setting it as 0.1 for DO2HSC due to the change of the interval area.
> > > >
> > > > \begin{matrix}
> > > > \hline
> > > > Method & Dataset/Percentile & 0.01 &0.1 &0.5& 0.8\\\\\hline
> > > > 		DOHSC & MUTAG & \textbf{0.9527(0.0187)} & 0.9497 (0.0249) & 0.9112 (0.0099) & 0.8675 (0.1287)\\\\
> > > > 		& COX2 & \textbf{0.6041(0.0661)} & 0.6022 (0.0789) & 0.5232 (0.0494) & 0.5523 (0.0572)\\\\
> > > >   \hline
> > > >   		Method & Dataset/Percentile & 0.01 &0.05 &0.1& - \\\\
> > > >     \hline
> > > > 		DO2HSC & MUTAG & \textbf{0.9089(0.0609)} &  0.8041 (0.1006)& 0.6769 (0.1207) & -\\\\
> > > > 		& COX2 & \textbf{0.6329(0.0292)} & 0.6149 (0.0187)& 0.5830 (0.0713)& -\\\\
> > > > 		\hline
> > > > \end{matrix}

---

> ### Author Response · Authors · 2022-12-12
> **Any feedback to our rebuttal?**
>
> Hi Reviewer,
>
> We haven't received any feedback from you. Your major concern is that our method is not specifically designed for graph data. We think this is not a limitation of our work. Our method is applicable to more general cases of anomaly detection. We have provided additional results on non-graph datasets, which showed that our methods have competitive performance compared to the state-of-the-art. More importantly, on graph datasets, our methods have promising performance.
> For your other concerns, we also provided explanations or numerical justification. Please take a look at them and feel free to comment. Thanks for your evaluation and time.
>
> Best regards,
>
> Authors

---

### Official Review · Reviewer_mmsV · 2022-11-03

**Confidence:** 4
**Correctness:** 3
**Technical Novelty And Significance:** 3
**Empirical Novelty And Significance:** 2
**Recommendation:** 6

**Clarity, Quality, Novelty And Reproducibility:**

The paper readability is okay.

As for the quality, there is a certain issue with weakness 1.
As for experiments see weakness 3.
There is no analysis why the two proposed changes are helpful.
These factors reduce the quality somewhat.

Regarding quality and reproducibility: several AUCs in the baseline experiments are substantially below 0.5. This indicates that one would obtain a very good AUC if one inverses the prediction rule for outliers. This can be due to the instability of the baselines or due to suboptimal parameters. One can choose not to hold it against the submission.

The two propositions over infograph+SVDD are novel.

**Strength And Weaknesses:**

Strengths:
the paper does several experiments on graph datasets.

Weaknesses:

1. the down projection does something else than the text states,

when applying W = V_{k'} \Lambda^{-1}_{k'}
to H= U \Lambda V^\top
then the result is:
U_{k'} (in math) aka U[:,:d]  (in python/pytorch)

It is NOT U[:,:d] I_d V^\top[:d,:] as what would be the orthogonal projection onto the first d dimensions of a SVD decomposition.

What did the authors want to compute  ?
 If it is U[:,:d], then the approach needs to be renamed, as that is not an orthogonal projection onto the first dimensions of U[:,:d] I_d V^\top[:d,:] .

2. It is not clear whether the singular vector standardization (by \Lambda^{-1}_{k'}) or the downprojection onto the first k' helps to improve DOHSC over infograph+SVDD.
To understand this, an experiment would be useful where one only projects onto the top k' dimensions in the Orthogonal_Projector module, without the .matmul(torch.linalg.inv(S)[:d,:d]) in the weightConstraint class  (or just .matmul(torch.eye(d))  )

3. the outlier detection is based on class-labels only. Depending on the dataset this might cluster too easily compared to outliers in the wild.

4. an analysis is missing why prohibiting too small distances from the center is a good idea.


**Summary Of The Paper:**

The authors deal with outlier detection on graphs.
They start with the paper on InfoGraph "INFOGRAPH: UNSUPERVISED AND SEMI-SUPERVISED
GRAPH-LEVEL REPRESENTATION LEARNING VIA MUTUAL INFORMATION MAXIMIZATION" by Sun et al , combined with SVDD, and propose two modifications.

1. DOHSC adds on top of infograph+SVDD a singular vector standardization and a projection on the first dimensions corresponding to the largest k' = 8 directions on the right side of the SVD (equation 9) . (see notes in Strength and weaknesses, that the code does actually something different)

2. DO2HSC replaces the SVDD objective by a two-sided objective which aims to identify outliers by samples which have a too large or too small distances from the center.

This comes with a 2 sided loss (penalizing too large and too small distances from the center), and calculates a distance measure which is positive if the sample distance from the center is either too large or too small.



**Summary Of The Review:**

The paper proposes two interesting changes. The orthogonal projection could be a mistaken implementation, see weakness 1, which is a point of lack of clarity and quality which has to be addressed.

*edit* after reading the rebuttal, the reviewer is agreeable to increase his rating to marginally above the accept threshold.

---

> ### Author Response · Authors · 2022-11-17
> **Thanks for the review. We added comparisons of the orthogonal projection layer with or w/o standardization.**
>
> **Response to questions:**
>
> **Q1: The down projection does something else than the text states, when applying $\mathbf{W} = \mathbf{V}\_{k'}\mathbf{\Lambda}^{-1}\_{k'}$ to $\mathbf{H}= \mathbf{U \Lambda V^\top}$ then the result is: $\mathbf{U}\_{k'}$ (in math) aka $\mathbf{U}[:,:d]$ (in python/pytorch). It is NOT $\mathbf{U}[:,:d] \mathbf{I}\_d \mathbf{V}^\top[:d,:]$ as what would be the orthogonal projection onto the first d dimensions of a SVD decomposition. What did the authors want to compute? If it is $\mathbf{U}[:,:d]$, then the approach needs to be renamed, as that is not an orthogonal projection onto the first dimensions of $\mathbf{U}[:,:d] \mathbf{I}\_d \mathbf{V}^\top[:d,:]$.**
>
> **Response:** Thanks for pointing out this confusion. In our paper, "orthogonal projection" is to find a projection matrix for $\mathbf{H}\in\mathbb{R}^{N\times k}$ such that the columns of the projected data $\mathbf{Z}:=\mathbf{HW}$ are orthogonal, not to compute the projection onto the column or row space of $\mathbf{H}$, though they are closely related to each other. Specifically, in our method, the SVD of $\mathbf{H}$ is $\mathbf{H}= \mathbf{U}\mathbf{\Lambda}\mathbf{V}^\top$. We let $\mathbf{W} := \mathbf{V}\_{k'}\mathbf{\Lambda}^{-1}\_{k'}$, then $\mathbf{Z}=\mathbf{HW}=\mathbf{U}\_{k'}$ and $\mathbf{Z}^\top\mathbf{Z}=\mathbf{I}\_{k'}$. This is equivalent to performing PCA and using the standardized principal components. The explanation is supplemented to our paper and we will change the name to "pursuing orthogonal (or uncorrelated) features" if the reviewer thinks it is necessary.
>
> **Q2: It is not clear whether the singular vector standardization (by $\Lambda^{-1}_{k'}$) or the downprojection onto the first $k'$ helps to improve DOHSC over infograph+SVDD. To understand this, an experiment would be useful where one only projects onto the top $k'$ dimensions in the Orthogonal Projector module, without the .matmul(torch.linalg.inv(S)[:d,:d]) in the weightConstraint class (or just .matmul(torch.eye(d))).**
>
> **Response:** Thanks for the insightful suggestion. We added the corresponding experiment as follows. The performance of DOHSC without standardization is acceptable and most results of it are better than those of the two-stage baseline, i.e., infoGraph+Deep SVDD. It verified the superiority of the end-to-end method over the two-stage baselines. However, the model with standardization outperformed the non-standardized one in almost all cases.
> \begin{matrix}
> \hline
>         & Class & infoGraph+Deep\\,SVDD  & DOHSC\\,(Non\text{-}Standardization) & DOHSC                 \\\\\hline
> MUTAG                & 0     & 0.8805 \pm 0.0448  & 0.8521 \pm 0.0650       & \textbf{0.8822 $\pm$ 0.0432}   \\\\
>                              & 1     & 0.6166 \pm 0.2052  & 0.6918 \pm 0.1467      & \textbf{0.8115 $\pm$ 0.0279}   \\\\\hline
> COX2       & 0     & 0.4825
> \pm 0.0624 & 0.5800 \pm 0.0473                          & \textbf{0.6263 $\pm$ 0.0333}   \\\\
>                              & 1     & 0.5029 \pm 0.0700  & 0.5029 \pm 0.0697                         &\textbf{0.6805  $\pm$ 0.0168}  \\\\\hline
> ER\\_MD      & 0     & 0.5312 \pm 0.1545  & 0.4881 \pm 0.0626                         & \textbf{0.6620   $\pm$ 0.0308} \\\\
>                              & 1     & 0.5682 \pm 0.0704  & \textbf{0.6740  $\pm$ 0.0356}                         & 0.5184 \pm 0.0793   \\\\\hline
> DD          & 0     & 0.3942 \pm 0.0436  & 0.4029 \pm 0.0354                         & \textbf{0.7083  $\pm$ 0.0188}  \\\\
>                              & 1     & 0.6484 \pm 0.0236  & 0.6903 \pm 0.0215                         & \textbf{0.7579  $\pm$ 0.0154}  \\\\\hline
> IMDB\text{-}Binary & 0     & 0.6353 \pm 0.0277  & 0.5149 \pm 0.0655                         & \textbf{0.6609 $\pm$ 0.0033}   \\\\
>                              & 1     & 0.5836 \pm 0.0995  & 0.6505 \pm 0.0585                        & \textbf{0.7705 $\pm$ 0.0045}   \\\\\hline
> COLLAB      & 0     & 0.5662 \pm 0.0597  & 0.6067 \pm 0.1007                         & \textbf{0.9185 $\pm$ 0.0455}  \\\\
>                              & 1     & 0.7926 \pm 0.0986  & 0.8958 \pm 0.0141                         & \textbf{0.9755 $\pm$ 0.0030}  \\\\
>                              & 2     & 0.4062 \pm 0.0978  & 0.4912 \pm 0.2000
>                          & \textbf{0.5450 $\pm$ 0.0469}\\\\ \hline
> \end{matrix}

---

> > ### Author Response · Authors · 2022-11-17
> > **Thanks for the review. We respond to your concern about one-class outlier detection.**
> >
> > **Q3: The outlier detection is based on class-labels only. Depending on the dataset this might cluster too easily compared to outliers in the wild.**
> >
> > **Response:** Thank you for your comments. First of all, a label-based outlier is commonly used in other outlier detection work [1-6], and we adopt a similar approach mainly for better comparison. Second, we want to claim that the proposed methods are designed in an unsupervised manner. None of the class-label is used in the training and testing stage. Finally, we try to give a wild-dataset-based anomaly detection experiment, which is specifically conducted on a reformed test dataset from different sources. However, it is regrettable that we haven't found any datasets that are specific to graph-level outlier detection. We expect to construct a dataset in future work and handle this problem.
> >
> > **Reference:**
> >
> > [1] Hanqiu Deng and Xingyu Li. Anomaly detection via reverse distillation from one-class embedding. In IEEE/CVF Conference on Computer Vision and Pattern Recognition, pp. 9727–9736, 2022.
> >
> > [2] Sachin Goyal, Aditi Raghunathan, Moksh Jain, Harsha Vardhan Simhadri, and Prateek Jain. DROCC: deep robust one-class classification. In Proceedings of the 37th International Conference on Machine Learning, pp. 3711–3721, 2020.
> >
> > [3] Rongrong Ma, Guansong Pang, Ling Chen, and Anton van den Hengel. Deep graph-level anomaly detection by glocal knowledge distillation. In Proceedings of the Fifteenth ACM International Conference on Web Search and Data Mining, pp. 704–714, 2022.
> >
> > [4] Chen Qiu, Marius Kloft, Stephan Mandt, and Maja Rudolph. Raising the bar in graph-level anomaly
> > detection. In Proceedings of the Thirty-First International Joint Conference on Artificial Intelligence, pp. 2196–2203, 2022.
> >
> > [5] Lukas Ruff, Robert Vandermeulen, Nico Goernitz, Lucas Deecke, Shoaib Ahmed Siddiqui, Alexander Binder, Emmanuel Muller, and Marius Kloft. Deep one-class classification. In International Conference on Machine Learning, pp. 4393–4402, 2018.
> >
> > [6] Jue Wang and Anoop Cherian. GODS: Generalized one-class discriminative subspaces for anomaly detection. In IEEE/CVF International Conference on Computer Vision, pp. 8200–8210, 2019.

---

> > > ### Author Response · Authors · 2022-11-17
> > > **Thanks for the review. We analyze our motivation for bi-hypersphere compression.**
> > >
> > > **Q4: An analysis is missing why prohibiting too small distances from the center is a good idea.**
> > >
> > > **Response:** Thanks for bringing up the discussion about the intuition behind bi-hypersphere compression. We came up with the bi-hypersphere approach mainly for the following two reasons.
> > >
> > > - First, empirically, we have found that anomalous data falling in the hypersphere are often close to the centroid especially when that region has few or even no training (normal) data points. In addition, the training data are often far from the centroid. Thus, we want to reduce the volume of the space enclosing the training data.
> > > - Second, the empirical findings are actually related to the characteristics of high-dimensional distributions. The anomaly score is determined by the $\ell_2$ norm $\Vert\mathbf{z}-\mathbf{c}\Vert$, where $\mathbf{c}$ denotes the centroid. Let $\tau$ be a threshold determined by a certain quantile such as 0.95 or 0.99. If $\Vert\mathbf{z}-\mathbf{c}\Vert\geq \tau$, then $\mathbf{z}$ is abnormal; otherwise, $\mathbf{z}$ is normal. However, most data points satisfy $\tau'\leq \Vert\mathbf{z}-\mathbf{c}\Vert$, where $\tau'$ is close to $\tau$ especially when the dimension is high. This indicates that within the range $[0,\tau']$, the normality is not supported by the training data. Take multivariate Gaussian simulation (Please see Figure 6 in Appendix B of our paper) as an example. Since the peak (mode) of the probability density function is at the origin, it is natural to expect most samples to be near the origin. However, in the high-dimensional cases, the typical set (where data has information closest to the expected entropy of the population) of a Gaussian is the thin shell within a distance from the origin. To be more precise, we show the following table, in which we consider multivariate Gaussians with different dimensions. We see that the higher the dimension is, the greater the quantities of data further away from the center.
> > > Actually, for any $d$-dimensional Gaussian $\mathcal{N}(\mathbf{0},\mathbf{I}_d)$, (using Lemma 1 of Laurent and Massart 2000), we can prove that
> > > $\mathbb{P}\left[\Vert \mathbf{z}\Vert \geq \sqrt{d-2\sqrt{dt}}\right] \geq 1-e^{-t}$,  for all $t > 0$, which matches the values in the following table.
> > >
> > > \begin{matrix}
> > > \hline
> > >  Quantile \\;(correspond\\;to\\;\tau' or ~ \tau) & dim=1  & dim=10 & dim=50 & dim=100 & dim=200 & dim=500 \\\\ \hline
> > > 0.01 & 0.0127 & 1.5957 & 5.5035 & 8.3817  & 12.5117 & 20.6978 \\\\
> > > 0.25 & 0.3115 & 2.5829 & 6.5380  & 9.4908  & 13.6247 & 21.8542 \\\\
> > > 0.50 & 0.6671 & 3.0504 & 7.0141 & 9.9662  & 14.1054 & 22.3337 \\\\
> > > 0.75 & 1.1471 & 3.5399 & 7.5032 & 10.4386 & 14.5949 & 22.8200   \\\\
> > > 0.99 & 2.5921 & 4.8265 & 8.7723 & 11.6049 & 15.7913 & 24.0245 \\\\\hline
> > > \end{matrix}
> > > In the table, when the dimension is higher, $\tau'$ is closer to $\tau$.
> > > The related explanations and numerical simulations are in Appendix B of our paper. We also give the following two references to support our thoughts.
> > >
> > > **References:**
> > >
> > > [1] Roman Vershynin. 2018. Similarity of normal and spherical distributions. In High-Dimensional Probability An Introduction with Applications in Data Science. Cambridge University Press, 53-53.
> > >
> > > [2] Mianzhi Wang. 2018. The Counterintuitive Behavior of High-Dimensional Gaussian. research.wmz.ninja. https://research.wmz.ninja/articles/2018/03/the-counterintuitive-behavior-of-high-dimensional-gaussian-distributions.html
> > >
> > > **Thank you again for all your comments, which have made our paper stronger.**

---

> > > > ### Comment · Reviewer_mmsV · 2022-12-04
> > > > **after reading the rebuttal**
> > > >
> > > > ... the reviewer is agreeable to increase his rating to marginally above the accept threshold.
> > > >
> > > > Reason:
> > > > The authors clarified the mistunderstanding, and performed a useful ablation experiment.
> > > >
> > > > Understanding why some outliers sometimes end up close to the centroid remains open. Also the slight weakness, that only label-based anomalies were tested.

---

> > > > > ### Author Response · Authors · 2022-12-04
> > > > > **We highly appreciate your feedback.**
> > > > >
> > > > > The authors sincerely thank you for evaluating our rebuttal and raising the score.

---

### Decision · Program_Chairs · 2023-01-20

**Decision:**

Reject

**Justification For Why Not Higher Score:**

The weaknesses with respect to empirical comparison are crucial and should be addressed. Together with the fact that the empirical performance of the proposal on AIDS and PROTEINS newly added in the authors' response are not convincing, this paper is not ready for publication at the moment. I think the paper potentially contains an interesting contribution, thereby I strongly recommend carefully re-organizing empirical evaluation and also addressing other weakness points.


**Justification For Why Not Lower Score:**

N/A

**Metareview: Summary, Strengths And Weaknesses:**

This paper addresses the problem of graph-level anomaly detection, where the task is to find anomalous graphs from a collection of graphs. The proposed method learns graph embedding based on GIN and mutual information maximization, followed by performing bi-hypersphere compression to emphasize discrimination of anomalous graphs from normal graphs. The performance of the proposal is empirically examined on real-world datasets.

### Strength

- The idea of bi-hypersphere compression with its combination to GNNs is interesting. I really expect further analysis in this direction.

- The quality of presentation is high. This paper is overall clearly written and easy to follow.

### Weakness

- As reviewers pointed out, the motivation and the proposed methodology do not match very well; that is, this study is motivated by graph-level anomaly detection, while the newly proposed method is not a graph-specific technique. This issue should be resolved.

- The bi-hypersphere compression needs more analysis. Although I appreciate the authors' response, currently it is not so clear why this leads to improvement of anomaly detection. Theoretical analysis would be desirable.

- Empirical comparison seems to be unfair. The authors mention that only normal data are available for a training dataset. However, in the fully unsupervised anomaly detection, this is not possible, and one needs to directly find anomalies from a dataset that contain both normal and abnormal data without any class labels. I think graph kernel based methods work in the latter setting, and this may be one of reasons for their inferior performance compared to GNN-based methods, although graph kernels, in particular the WL kernel, are known to be often empirically competitive to GNNs in many cases.

- Comparison is also unfair with respect to parameter tuning. Since the parameter tuning is fundamentally difficult in unsupervised learning, it should be carefully considered in experiments for fair comparison. I think the empirically optimal parameters are chosen in the proposal, then this should be applied to all the comparison partners. Please clearly state how to tune parameters for all the comparison partners. In particular, in the WL kernel, different numbers of iterations are just tested and the optimal one is not chosen. So please just present the optimal performance as the WL's result, which actually reduces the space of the paper and the authors can add more methods.
In addition, although I acknowledge an additional experiment added in the authors' response about the sensitivity, it is not convincing as the most important part is missing. In this table, there is a monotonic trend that smaller $\nu$ leads to better results. Does this mean that $\nu = 0$ leads to the best result? I do not think so, and then it is important to examine when the performance drops when $\nu$ decreases.

- There are other issues in experiments raised by the reviewers, e.g., lack of comparison with recent methods.